# ThreadWeaver: Adaptive Threading for Efficient Parallel Reasoning in Language Models

**Long Lian** [1 2]  **Sida Wang** [1]  **Felix Juefei-Xu** [1]  **Tsu-Jui Fu** [1]  **Xiuyu Li** [2]  **Adam Yala** [2 3]  **Trevor Darrell** [2]
**Alane Suhr** [2]  **Yuandong Tian** [1]  **Xi Victoria Lin** [1]

## Abstract

Scaling inference-time computation has enabled Large Language Models (LLMs) to achieve strong reasoning performance, but their inherently sequential decoding incurs substantial latency, motivating parallelization of the generation process. However, existing parallel reasoning approaches suffer from performance degradation compared to their sequential counterparts, and often rely on specialized inference engines. We introduce ThreadWeaver, a framework for adaptive parallel reasoning that matches the accuracy of comparably sized sequential reasoning models while significantly reducing inference latency via three key innovations: **1)** a two-stage parallel trajectory generator that produces high-quality parallel chain-of-thought data for supervised fine-tuning; **2)** a trie-based rollout design that enables parallel reasoning on any off-the-shelf autoregressive inference engine; and **3)** a parallelization-aware reinforcement learning framework that trains the model to balance reasoning accuracy with effective parallelization. Across six challenging math reasoning benchmarks, ThreadWeaver trained on top of Qwen3-8B achieves performance on par with cutting-edge sequential reasoning models (**79.9%** on AIME24 and **71.9%** on average) while delivering up to **1.53×** speedup in token latency, establishing a new Pareto frontier between accuracy and efficiency.

[1]Meta Superintelligence Labs (MSL), Menlo Park, CA, United States [2]UC Berkeley, Berkeley, CA, United States [3]UCSF, San Francisco, CA, United States. Correspondence to: Long Lian <longlian@berkeley.edu>, Xi Victoria Lin <victoriaxlin.ai@gmail.com>.

*Proceedings of the 43rd International Conference on Machine Learning*, Seoul, South Korea. PMLR 306, 2026. Copyright 2026 by the author(s).

## 1 Introduction

Large Language Models (LLMs) have advanced automated problem solving across domains that require complex reasoning, including mathematics, programming and scientific discovery. This process is driven by scaling inference-time computation to produce longer and more detailed chain of thought (CoT) that iteratively navigates the problem space and revises earlier steps (Wei et al., 2022; Zelikman et al., 2022a; Yao et al., 2023). As task difficulty increases, the time required to reach a correct solution rises dramatically, in some cases reaching tens of hours for a single task (Anthropic, 2025). On the other hand, the sequential nature of autoregressive LLMs imposes severe inference latency which cannot be easily reduced by adding more compute resources, as generating each token requires conditioning on all previous tokens.

Parallel reasoning has since emerged as a popular technique for mitigating this limitation in LLMs, offering an additional axis for test-time scaling beyond extending sequence length. Methods such as best-of-N and self-consistency (Wang et al., 2023) execute multiple independent reasoning threads and aggregate their outputs. Test-time search approaches such as tree-of-thought (Yao et al., 2023) guide exploration through hand-crafted structures tuned to the underlying problem. More recent works have explored *adaptive parallel reasoning*, where LLMs are trained to decide when to branch reasoning into parallel threads and when to continue sequentially, mimicking multi-threaded execution without predefining the branching structure (Pan et al., 2025; Yang et al., 2025b; Jin et al., 2025; Zheng et al., 2025). Such adaptive approaches hold the promise of allocating test-time computation more effectively across reasoning stages, enabling the model to self-determine when parallelization is beneficial for the task at hand.

However, despite its potential, *adaptive parallelization has yet to demonstrate competitive performance on complex real-world problems comparable to that of strong sequential reasoning models* (Yang et al., 2025b; Zheng et al., 2025), due to several fundamental challenges. First, high-quality parallel reasoning trajectories for real-world problems are difficult to obtain for training purposes, leaving

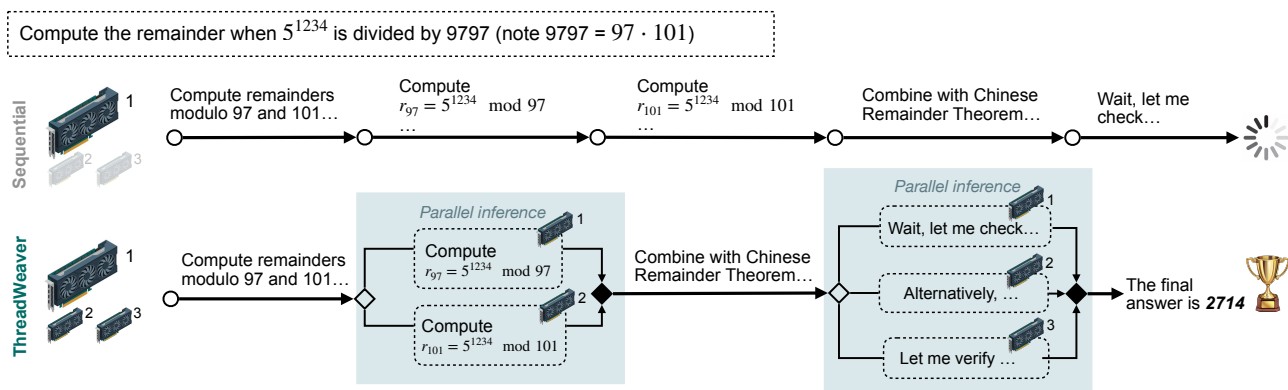

*Figure 1.* For complex reasoning problems, a sequential reasoner generates tokens one at a time, so latency scales with the trajectory length and is difficult to reduce with additional compute. ThreadWeaver instead adaptively spawns ($\diamond$) and joins ($\blacklozenge$) concurrent threads that work on different parts of the solution, using extra compute to shorten the critical reasoning path.

prior datasets limited in both scale and diversity. Second, most existing approaches require modifications to position embeddings, key-value caches, or attention masks at inference to support thread branching and joining (Jin et al., 2025; Zheng et al., 2025; Yang et al., 2025b), making deployment on standard inference engines difficult (Kwon et al., 2023; Zheng et al., 2024). Third, scaling reinforcement learning (RL) for parallel reasoning is further complicated by additional modeling and system challenges, such as advantage calculation across branches and maintaining consistency between train-time rollouts and test-time execution.

To address these challenges, we propose **ThreadWeaver**[1], a framework that brings RL-induced adaptive parallel reasoning to strong reasoning LLMs such as Qwen3 (Yang et al., 2025a), while requiring no modifications to the underlying inference engine. We focus on a setting where abundant compute is available to reduce reasoning latency on problems that require long chain-of-thoughts. For such examples, further latency reduction requires parallelism *within* a single reasoning trajectory. ThreadWeaver is designed specifically for this regime and introduces three core innovations: *1) a two-stage parallel trajectory generator*, which generates high-quality structured reasoning traces with LLM-annotated parallelization control tokens, providing a scalable, low-cost cold start for downstream RL (§3); *2) a trie-based training and inference co-design* (§2), which enables parallel reasoning on standard autoregressive inference engines such as vLLM (Kwon et al., 2023) and SGLang (Zheng et al., 2024); *3) parallelization-aware GRPO (P-GRPO)*, which jointly optimizes accuracy and latency reduction through mathematically justified thread-wise advantage broadcast (§4.1) and a parallelization-aware reward design (§4.2) to induce effective adaptive parallelization while maintaining strong reasoning performance.

**ThreadWeaver is the first adaptive parallel reason-**

ing framework that achieves accuracy comparable to equally sized cutting-edge sequential reasoning models (79.9% accuracy for ThreadWeaver vs 78.3% for the thinking model Qwen3-8B (Yang et al., 2025a) on AIME24, and 71.9% vs 72.2% on average across six widely used math benchmarks) while delivering substantial reductions in token latency: 1.14× speedup on AIME24 (Mathematical Association of America, 2024), 1.16× speedup on AMC23 (Mathematical Association of America, 2023), 1.23× speedup on MATH500 (Hendrycks et al., 2021), 1.21× on OlympiadBench (He et al., 2024), and 1.53× on Minerva Math (Lewkowycz et al., 2022). These results establish a new Pareto frontier between accuracy and efficiency, opening a promising direction for enabling LLMs to tackle increasingly complex problems within practical inference time budget.

## 2 Model

### 2.1 Parallel Trajectory Format

We extend standard autoregressive (AR) generation with lightweight control tokens arranged in a fork-join pattern to realize adaptive parallel reasoning, following Multiverse (Yang et al., 2025b). Unlike Multiverse, our framework restricts trajectories to a single level of parallelization in which all branches rejoin the main thread.[2] Below we describe our parallel trajectory format, with an illustrative example shown in Figure 2:

- `<think>` and `</think>` mark the start and end of the reasoning trajectory, which may contain sequential segments and zero or more parallel blocks.

---

[1]Code: https://github.com/facebookresearch/ThreadWeaver

[2]In practice, we found that single-level parallelization often offers significant speedup while preserving reasoning accuracy. It also reduces the engineering overhead and data-quality challenges introduced by nested parallelization branches. The techniques proposed in our work naturally generalize to nested branches by permitting parallel blocks within individual threads.

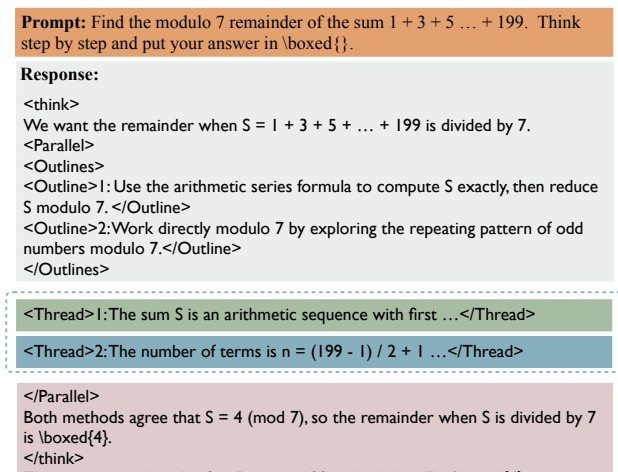

**Prompt:** Find the modulo 7 remainder of the sum 1 + 3 + 5 … + 199. Think step by step and put your answer in \boxed{}.

**Response:**

<think>
We want the remainder when S = 1 + 3 + 5 + … + 199 is divided by 7.
<Parallel>
<Outlines>
<Outline>1: Use the arithmetic series formula to compute S exactly, then reduce S modulo 7. </Outline>
<Outline>2:Work directly modulo 7 by exploring the repeating pattern of odd numbers modulo 7.</Outline>
</Outlines>

<Thread>1:The sum S is an arithmetic sequence with first …</Thread>
<Thread>2:The number of terms is n = (199 - 1) / 2 + 1 …</Thread>

</Parallel>
Both methods agree that S = 4 (mod 7), so the remainder when S is divided by 7 is \boxed{4}.
</think>
The remainder when 1 + 3 + 5 + … + 199 is divided by 7 is \boxed{4}.

*Figure 2.* **Format of our parallel reasoning trajectories.** <Parallel> encloses a fork-join block with <Outlines> and multiple <Thread>'s; content inside each <Thread> is generated *concurrently*, while the other spans are generated sequentially.

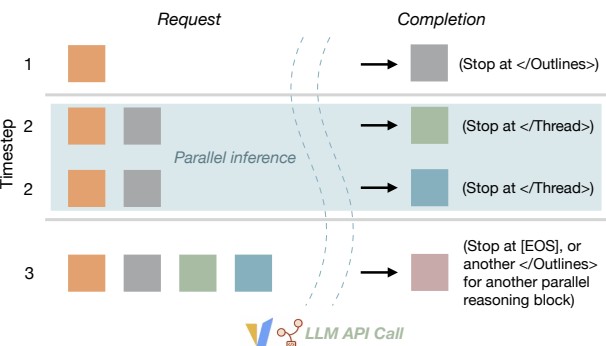

*Figure 3.* **Inference-time request sequence for a parallel reasoning trajectory.** For a given user prompt, Timestep 1 decodes the prefix and the <Outlines> block sequentially up to </Outlines>. Timestep 2 launches one completion request per outline in parallel, each seeded with its corresponding "<Thread> $i$:" prefix and stopped at </Thread>. Timestep 3 resumes sequential decoding over the joined context until [EOS] or the next </Outlines>. All steps are executed on a standard LLM inference engine.

- <Parallel>...</Parallel>: defines a parallel block.
- <Outlines>...</Outlines>: consists of numbered <Outline> entries which declare independent sub-tasks.
- <Thread> $i$: execution trajectory of the i-th sub-task. *Threads are generated independently and must not reference other threads.* At inference, each thread is generated in parallel and stopped at </Thread>.

In data collection, this format is enforced by our data generator proposed in Section 3 for supervised fine-tuning. During RL rollout and at inference time, this structured trajectory is generated by an orchestrator calling an LLM inference engine, described in Section 2.2.

## 2.2 Parallel Inference

Our parallel structure introduces <Parallel> blocks that contain an <Outlines> section followed by multiple <Thread>'s. Importantly, *only the thread contents are generated in parallel*, while all remaining parts of the trajectory are generated via standard autoregressive (AR) decoding. As shown in Figure 3, this design allows parallel inference to be implemented on top of any model exposed through a request-completion interface, such as an LLM inference server with API backed by an autoregressive inference engine like SGLang (Zheng et al., 2024) or vLLM (Kwon et al., 2023), without any modifications to their underlying system.

**State-machine view.** We implement inference as a minimal state machine operating on request-response pairs with a text-completion API. The orchestrator alternates between sequential phases (decoding until </Outlines>), parallel phases (launching independent completion requests for each thread until </Thread>), and join phases (concatenating results and continuing). This design requires no modifications to the LLM architecture or inference engine, inheriting existing optimizations like prefix caching and paged attention (Zheng et al., 2024; Kwon et al., 2023). Full state-machine specification is provided in Appendix C.

**Hybrid operation.** This design naturally enables hybrid reasoning: operators may choose *parallel mode* (executing the state machine for low latency via parallel reasoning) or *autoregressive mode* (bypassing the state machine for high throughput). ThreadWeaver is the first adaptive framework maintaining sequential reasoning compatibility.

## 2.3 Efficient Training with Trie-based Sequence Merging

We fine-tune the LLM to emit these parallel structures by matching training targets to the inference state machine via trie construction and traversal.

**Trie-based loss computation.** As shown in Figure 4, using regex-based parsing, we first extract all the ⟨context, completion⟩ units that will be issued during inference. Specifically, each context corresponds to a fragment issued by the state machine to a text completion API, and each completion corresponds to the LLM-generated span between structural tokens (e.g., from <think> to </Outlines>). The pair of spans is concatenated into a token sequence. We then insert all units into a token-level prefix tree (root = shared prompt, branches = divergent continuations), then use depth-first traversal to linearize it into a single training sequence with an *ancestor-only* attention mask and position

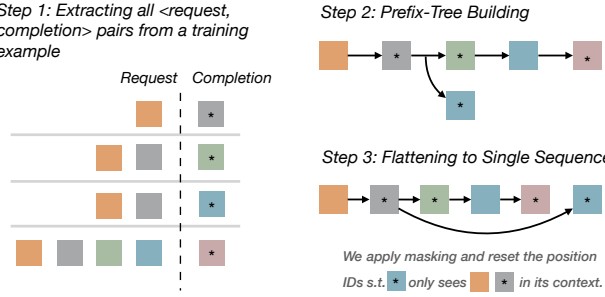

*Step 1: Extracting all <request, completion> pairs from a training example*

Request Completion

*Step 2: Prefix-Tree Building*

*Step 3: Flattening to Single Sequence*

*We apply masking and reset the position IDs s.t. ■ only sees* ■ ■ *in its context.*

*Figure 4.* **Our trie-based training sequence construction has three steps: 1)** extract context and completion segments that the inference state machine will encounter in order to produce this trajectory; **2)** insert these segments into a token-level trie (prefix-tree); **3)** traverse the trie to produce a flat sequence with an ancestor-only attention mask and standard incremental positions counting from the root node. ∗ denotes text spans with loss applied. Although we only show one branching in this example, ThreadWeaver supports branching and joining multiple times in a trajectory.

encoding to prevent cross-thread leakage.[3] Loss is applied only to tokens from the completion part of each unit. This ensures training matches the inference state machine's isolation property. Implementation is detailed in Appendix C.

**Consistency and scalability.** This co-design guarantees that all the tokens in a trajectory generated by the trie traversal will have the same context and position encodings as in each request-response pair that the orchestrator issues. Training with trajectories constructed in this manner also ensures that inferencing without our orchestrator recovers ordinary autoregressive reasoning behavior, since the overall trajectory obtained by autoregressive inferencing is always a valid subsequence of the trajectory obtained from the trie. We employ FlexAttention (Dong et al., 2024) for efficient attention masking in training.

## 3 Parallel Trajectory Curation for Supervised Fine-tuning

With our proposed inference pattern and training paradigm, a key challenge is to obtain suitable training data for supervised fine-tuning. Our training procedure requires parallel reasoning trajectories in the fork-join format described in Section 2.1. To construct these trajectories, we employ a two-stage pipeline: **1)** an LLM-driven annotation and a light-weight rewriting phase for SFT cold start, followed by **2)** self-training with reward-based filtering to scale data quality and improve performance.

**LLM rewriting.** We first extract sequential reasoning trajectories by running inference on Qwen3-8B (Yang et al., 2025a) with a dataset of prompts. We then apply a five-

---

[3]The change in attention masks and position encoding are only at training time and are not employed at inference time.

step LLM-driven annotation pipeline using GPT-5 (OpenAI, 2025) to identify parallelizable spans, extract canonical thread boundaries, rewrite for clarity and independence, generate outlines, and perform format validation. Crucially, we let the LLM recover structure from existing chain-of-thoughts by using line-based extraction, compared to re-generating solutions (Yang et al., 2025b), preserving the model's original reasoning semantics through minimal localized edits. Using this pipeline on 53k sequential Qwen3-8B trajectories from Polaris-53k (An et al., 2025), we obtain 959 high-quality parallelized trajectories for cold-start training. Full pipeline details and prompts are in Appendix F.

**Self-training with reward-based filtering.** While effective for cold-start, the LLM-rewritten trajectories produced in the previous step reflect post-hoc decomposition rather than the model's natural generative patterns. To bridge this gap, we fine-tune Qwen3-8B on the 959 cold-start items, then run parallel rollout on the full 53k prompts. We filter trajectories by format correctness and answer correctness, yielding a large corpus of format-faithful trajectories that align supervision with the model's own generative patterns, which serves as high-quality SFT data to enable reliable parallel rollout in subsequent reinforcement learning.

## 4 Reinforcement Learning

We build on Group Relative Policy Optimization (GRPO) (Shao et al., 2024), a policy-gradient method that computes advantages via intra-prompt normalization. For a prompt $p$ with $k$ sampled trajectories having rewards $\{r_1, \ldots, r_k\}$, let $\mu_p = \frac{1}{k} \sum_{j=1}^{k} r_j$ and $\sigma_p = \sqrt{\frac{1}{k} \sum_{j=1}^{k} (r_j - \mu_p)^2}$ denote the group mean and standard deviation. The group-normalized advantage is

$$A_{p,i}^{\text{GRPO}} = \frac{r_i - \mu_p}{\sigma_p + \varepsilon}, \tag{1}$$

with small $\varepsilon > 0$ for stability. We provide a detailed derivation of GRPO and its connection to our parallel extension in Section A.3.

### 4.1 P-GRPO: GRPO with Parallel Rollout

**State-machine based rollout.** We perform rollout using the inference orchestrator described in Section 2.2. Let $\tau_p^{(i)}$ be composed of the ordered set of ⟨context, completion⟩ units emitted by the state machine, $\tau_p^{(i)} = \{u_p^{(i,1)}, u_p^{(i,2)}, \ldots, u_p^{(i,M_i)}\}$, where $u_p^{(i,m)} = [\text{cont}_p^{(i,m)} \| \text{comp}_p^{(i,m)}]$ is the concatenation of the context $\text{cont}_p^{(i,m)}$ followed by its corresponding completion $\text{comp}_p^{(i,m)}$. Training with these units ensures the distribution seen during RL matches test-time execution so that there is no train–inference gap.

---

**Algorithm 1** Parallelization-Aware GRPO (P-GRPO)

1: **Input:** Policy $\pi_\theta$, batch of prompts $\mathcal{B}$, rollouts per prompt $k$, reward function $r(\cdot)$
2: **Output:** Updated policy $\pi_\theta$
3: **for** each prompt $p \in \mathcal{B}$ **do**
4:     **for** $i = 1$ to $k$ **do**
5:         Sample trajectory $\tau_p^{(i)}$ using parallel inference state machine (§2.2)
6:         Extract $\{u_p^{(i,1)}, \ldots, u_p^{(i,M_i)}\}$ where $u_p^{(i,m)} = [\mathrm{cont}_p^{(i,m)} \| \mathrm{comp}_p^{(i,m)}]$
7:         Compute reward $r_{p,i} = \mathcal{R}_{\mathrm{correct}}(\tau_p^{(i)}) + \mathcal{R}_{\mathrm{accel}}(\tau_p^{(i)})$
8:     **end for**
9:     Compute group mean: $\mu_p = \frac{1}{k} \sum_{j=1}^k r_{p,j}$
10:    **for** $i = 1$ to $k$ **do**
11:        Compute advantage: $A_{p,i} = r_{p,i} - \mu_p$
12:    **end for**
13: **end for**
14: Compute total tokens: $Z = \sum_{p \in \mathcal{B}} \sum_{i=1}^k T_{p,i}$
15: Compute loss: $\mathcal{L} = -\frac{1}{Z} \sum_{p,i} A_{p,i} \sum_{m=1}^{M_i} \log \pi_\theta(c_p^{(i,m)} \mid x_p^{(i,m)})$
16: Update $\theta$ by gradient descent on $\mathcal{L}$

---

**Advantage calculation.** We compute a single scalar reward $r_{p,i}$ for the *entire* trajectory (see Section 4.2) and *broadcast* its group-normalized advantage $A_{p,i}$ to all units in that sample. Let $T_{p,i}$ be the number of tokens in trajectory $i$ of prompt $p$ and $Z = \sum_{p \in \mathcal{B}} \sum_{i=1}^k T_{p,i}$ be the total token count. The loss is

$$\mathcal{L}_{\text{P-GRPO}}(\theta) = -\frac{1}{Z} \sum_{p,i} A_{p,i} \sum_{m=1}^{M_i} \log \pi_\theta\big(c_p^{(i,m)} \mid x_p^{(i,m)}\big), \tag{2}$$

where $c_p^{(i,m)}$ and $x_p^{(i,m)}$ denote completion and context tokens. Following the practice in sequential reasoning and multi-turn RL, we do not apply loss on the context tokens of each thread. Specifically, Equation (2) is implemented with a token-level mask that zeros out the loss on context tokens, so the log-probability term only accumulates contributions from completion tokens. Therefore, each generated token receives loss at most once, even if it later reappears as part of the context for subsequent generation.

This broadcasting across the units spanning all the parallel threads is mathematically justified, keeps the implementation simple, and empirically stabilizes training without per-branch credit heuristics. We provide derivations of P-GRPO and mathematical justification for thread-wise broadcast in Appendix B.

## 4.2 Parallelization-Aware Reward

Similar to sequential generation, we set the main reward to an indicator function of answer correctness. In addition, we softly encourage *acceleration* when parallelization reduces the critical path. For sample $\tau$, the total reward includes the two components added together:

$$r(\tau) = \mathcal{R}_{\mathrm{correct}}(\tau) + \mathcal{R}_{\mathrm{accel}}(\tau), \tag{3}$$

The correctness reward $\mathcal{R}_{\mathrm{correct}}(\tau)$ evaluates the boxed final answer:

$$\mathcal{R}_{\mathrm{correct}}(\tau) = \mathbf{1}\{\mathrm{CORRECT}(\tau)\}. \tag{4}$$

The acceleration reward $\mathcal{R}_{\mathrm{accel}}$ incentivizes the model to generate more efficient reasoning paths by leveraging parallelization:

$$\mathcal{R}_{\mathrm{accel}}(\mathbf{s}) = \mathbf{1}\{\mathrm{CORRECT}(\tau)\} \, \min\left(\rho \cdot \eta(\mathbf{s}), \rho_{\mathrm{clip}}\right) \tag{5}$$

where $\mathbf{s}$ denotes the set of parallel threads in the trajectory, $\rho$ is an acceleration reward scale, and $\rho_{\mathrm{clip}}$ is a clipping threshold.[4] Acceleration ratio $\eta(\mathbf{s})$ is computed as:

$$\eta(\mathbf{s}) = 1 - \frac{L_{\mathrm{longest}}}{L_{\mathrm{total}}} \tag{6}$$

where $L_{\mathrm{longest}}$ is the sequence length for the longest thread (i.e., token latency), $L_{\mathrm{total}}$ is the total number of tokens in the entire trajectory. The acceleration ratio reward is only given when the answer is correct, which is indicated by the correctness term in Equation (5).

**Removing normalization with standard deviation for stability.** Empirically, standard-deviation normalization in GRPO leads to unstable training with the multi-term reward Equation (3): when all rollouts are correct, mean-centering removes the correctness signal and std-dev normalization cancels the acceleration scale, causing the model to optimize acceleration at the expense of correctness. We instead use simple mean-centering, preserving the intended reward balance:

$$A_{p,i}^{\text{P-GRPO}} = r_{p,i} - \mu_p. \tag{7}$$

Full ablation analysis in Table 4. This choice is aligned with the modifications proposed by Liu et al. (2025), though our motivation originates from the multi-term reward structure proposed for incentivizing parallel reasoning.

Overall, since the rollout uses our parallel inference scheme specified in Section 2.2, P-GRPO learns parallel behavior that transfers directly to any off-the-shelf AR engine, along with correctness and acceleration improvements. We present the pseudo-code for our proposed P-GRPO in Algorithm 1.

---

[4]In our experiments we set $\rho = 0.5$ and $\rho_{\mathrm{clip}} = 0.2$. These values are chosen empirically to keep the acceleration bonus a small fraction of the correctness reward so that the model does not sacrifice accuracy for acceleration gains.

# 5 Experiments

## 5.1 Setup

We evaluate ThreadWeaver trained on top of Qwen3-8B (Yang et al., 2025a), a thinking model that demonstrates strong performance on mathematical problem solving with long CoT.

ThreadWeaver is trained through a three-stage pipeline: **1)** Fine-tuning Qwen3-8B on 959 LLM-rewritten parallel trajectories for 8 epochs, **2)** Self-training with 17,491 model-generated correct and structurally-valid trajectories, and **3)** RL using P-GRPO for 350 steps with batch size 128 and group size 8. Training prompts from all stages come from An et al. (2025). Sequential RL baseline uses the same training setup. Full details in Appendix C.

**Evaluation.** We evaluate performance on six math reasoning benchmarks: AIME24 (Mathematical Association of America, 2024), AIME25 (Mathematical Association of America, 2025), Minerva Math (Lewkowycz et al., 2022), AMC23 (Mathematical Association of America, 2023), OlympiadBench (He et al., 2024), and MATH500 (Hendrycks et al., 2021). Parallel reasoning in inference is performed via our proposed orchestrator.

## 5.2 Main Results

We report the performance of ThreadWeaver in Table 1, comparing it against a sequential GRPO baseline on Qwen3-8B using the same RL prompts across six math benchmarks. Both models are trained with identical RL configurations and evaluated with multiple rollouts per problem.

We report accuracy and token latency to assess reasoning correctness and efficiency. To quantify efficiency gains, we define three speedup metrics based on the ratio of token latencies between the sequential GRPO baseline and ThreadWeaver. *Speedup* is the average of these per-problem ratios across a dataset. *Speedup (correct only)* restricts this calculation to rollouts yielding correct final answers for both models. Finally, *Max Speedup (correct only)* identifies the maximum per-problem speedup achieved among correct rollouts, serving as an empirical upper bound for our method's acceleration. For each problem, we average token latency over all rollouts.

The *Speedup* row should be interpreted as the expected gain when sampling problems uniformly. A more conservative aggregate is the ratio of mean token latencies, which weights longer problems more heavily: $15.1k/13.2k = 1.14\times$. Excluding the benchmarks with the highest and lowest mean latencies gives a trimmed ratio-of-means of $1.20\times$. We report these views together because they answer complementary questions about per-problem acceleration and workload-level latency reduction.

**On average, ThreadWeaver matches the baseline's accuracy (71.9% vs. 72.2%) while reducing mean token latency from 15.1k to 13.2k and yielding up to $1.53\times$ speedup.**

Speedup depends not only on trace length, but also on problem decomposability and model familiarity with the task distribution. For example, AIME25 has long traces but remains close to parity, while Minerva Math has shorter traces yet larger speedup because many examples contain independent sub-computations or verification steps.

## 5.3 Comparisons

We compare ThreadWeaver with recently proposed adaptive parallel reasoning methods on AIME24, focusing on Multiverse (Yang et al., 2025b) and Parallel-R1 (Zheng et al., 2025). As summarized in Table 2, ThreadWeaver attains substantially higher accuracy (79.9%) compared to Multiverse (up to 53.8%) and Parallel-R1 variants (up to 19.4%). Furthermore, ThreadWeaver achieves higher *self-parallelism speedup* and *activation ratio*. The former is the ratio between the total number of tokens and the token latency in reasoning trajectory, and the latter is defined as the fraction of trajectories with parallelization. **ThreadWeaver achieves both higher accuracy and higher efficiency, establishing a stronger Pareto frontier than prior adaptive parallel reasoning approaches.**

We also observed that ThreadWeaver built on Qwen3-8B outperforms Multiverse, which is built on Qwen2.5-32B (Qwen et al., 2025), both in accuracy and in measured self-parallelism speedup. This indicates that our training recipe effectively induces adaptive parallelization behavior of the model at a much smaller model size. Compared to Parallel-R1, which is also based on a model from the Qwen3 family (Qwen3-4B-Base), ThreadWeaver operates in a long chain-of-thought regime that closely matches the sequence lengths of strong sequential reasoning models in the Qwen3 family. In contrast, Parallel-R1 is trained to produce significantly shorter chains of thought, which limits its test-time scaling ability and accuracy.

## 5.4 Acceleration Distribution Analysis

To analyze how acceleration varies across problems, Figure 5 shows per-problem speedup distributions on four math benchmarks. Speedup is defined as the ratio between the token latency of the sequential baseline and that of ThreadWeaver. For AIME24/AMC23/OlympiadBench, we average ratios over 32/8/8 rollouts per prompt to reduce sampling noise, while MATH500 uses a single rollout per problem.

Across all datasets, most mass lies to the right of the $1.0\times$ line, indicating consistent acceleration, with a wide spread reflecting strong question dependence. Some problems ad-

*Table 1.* Comparison between the sequential GRPO baseline and ThreadWeaver on Qwen3-8B across six math benchmarks. ThreadWeaver matches the baseline's accuracy (71.9% vs. 72.2%) while **reducing mean critical-path length from 15.1k to 13.2k tokens** and achieving **up to 1.53× average speedup** and **up to 3.56× max speedup**. Speedups are computed as ratios of the token latencies of the sequential reasoning baseline and ThreadWeaver.

| Model | AIME24 (Avg@32) | AIME25 (Avg@32) | AMC23 (Avg@8) | MATH500 (Avg@1) | Minerva Math (Avg@4) | OlympiadBench (Avg@8) | Average |
|---|---|---|---|---|---|---|---|
| **Accuracy (%)** | | | | | | | |
| Qwen3-8B + Seq. RL | 78.3 | 61.6 | 92.6 | 91.8 | 43.9 | 65.0 | 72.2 |
| Qwen3-8B + ThreadWeaver | 79.9 | 60.5 | 92.3 | 91.4 | 43.7 | 63.5 | 71.9 |
| **Token Latency (tokens)** | | | | | | | |
| Qwen3-8B + Seq. RL | 19.4k | 24.6k | 13.8k | 7.2k | 10.6k | 15.2k | 15.1k |
| Qwen3-8B + ThreadWeaver | **16.9k** | **24.0k** | **12.0k** | **6.4k** | **7.3k** | **12.8k** | **13.2k** |
| **Speedup** | 1.14× | 1.03× | 1.16× | 1.23× | 1.53× | 1.21× | 1.22× |
| **Speedup (correct only)** | 1.14× | 1.05× | 1.17× | 1.23× | 1.47× | 1.20× | 1.21× |
| **Max Speedup (correct only)** | 1.47× | 1.21× | 1.67× | 3.05× | 3.56× | 1.92× | – |

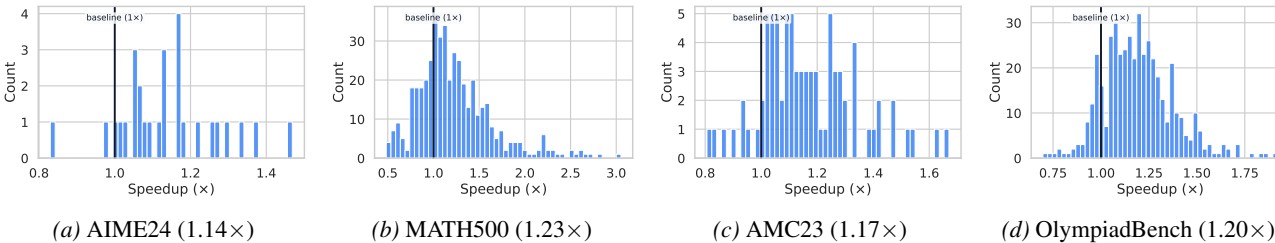

| *(a)* AIME24 (1.14×) | *(b)* MATH500 (1.23×) | *(c)* AMC23 (1.17×) | *(d)* OlympiadBench (1.20×) |
|---|---|---|---|

*Figure 5.* **Per-problem speedup distributions on math benchmarks.** The speedup is the ratio between the token latency of the sequential reasoning baseline and the token latency of ThreadWeaver. A vertical reference line at 1.0× marks parity with the sequential baseline. Parentheses show mean speedup per dataset.

*Table 2.* Comparison with other adaptive parallel reasoning approaches on AIME24. Self-parallelism speedup is the ratio between the total number of tokens and the critical path length. Activation ratio is defined as the percentage of samples with parallelization in the inference trajectories. ThreadWeaver achieves higher accuracy and stronger self-parallelism than Multiverse and Parallel-R1. Results from Yang et al. (2025b); Zheng et al. (2025).

| Model | Size | Self-Par. Speedup | Act. Ratio | Acc. |
|---|---|---|---|---|
| Multiverse-zero | 32B | 1.04× | - | 52.1% |
| Multiverse | 32B | 1.18× | - | 53.8% |
| Parallel-R1-Seen | 4B | - | 27.3 | 19.4% |
| Parallel-R1-Unseen (S1) | 4B | - | 13.6 | 18.3% |
| Parallel-R1-Unseen (S2) | 4B | - | 63.0 | 16.3% |
| **ThreadWeaver (ours)** | 8B | **1.25×** | **85.2** | **79.9%** |

*Table 3.* Wall-clock latency on 50 MATH problems (averaged over 8 runs per problem).

| Parallelization | Avg Latency (s) |
|---|---|
| Parallel | 142.21 |
| Sequential | 162.34 |
| Speedup | 1.14× |

mit large gains (up to 1.47× on AIME24 and 3.05× on MATH500), while others show little benefit or mild slowdowns. Overall, ThreadWeaver yields substantial critical-path reductions on decomposable problems and naturally approaches parity on intrinsically sequential ones.

This behavior is consistent with Amdahl's law: the sequential portion of the reasoning trace, branch imbalance, and orchestration overhead all bound achievable acceleration. Thus, ThreadWeaver should be viewed as improving the speed-accuracy Pareto frontier for problems with exploitable parallel structure, rather than as providing universal acceleration for intrinsically sequential tasks.

### 5.5 Ablations

**Adding compute makes ThreadWeaver faster.** To verify that critical-path reductions yield real-world gains, we measured wall-clock latency on 50 problems from MATH500 (8-run average). We compared standard autoregressive decoding on a single GPU against the orchestrator distributing threads across four GPUs at batch size 1.

As shown in Table 3, parallelization achieves a 1.14× speedup. While lower than token-latency gains due to system overhead, this confirms that ThreadWeaver effectively converts additional compute into reduced inference latency.

**Removing standard-deviation normalization in GRPO improves stability.** We compared GRPO variance normal-

*Table 4.* Effect of std-dev normalization on AIME 24 (Avg@32).

| Setting | Acc. | Seq. Len. | Longest |
|---|---|---|---|
| With Std. Norm. | 74.8% | 30.1k | 18.7k |
| Mean-Centered (Ours) | **79.9%** | **21.1k** | **16.9k** |

*Table 5.* Ablation on parallel rollout and self-training (AIME 24).

| Par. Rollout | Self-Train | Acc. | Latency |
|---|---|---|---|
| ✓ | ✗ | 77.9% | 17.6k |
| ✗ | ✓ | 78.4% | 17.3k |
| ✓ | ✓ | **79.9%** | **16.9k** |

ization against simple mean-centering. As shown in Table 4, mean-centering preserves the intended balance between correctness and acceleration, resulting in higher accuracy and shorter critical paths.

**Parallel rollout and self-training improve performance.** We ablate two training components: *parallel rollout* during RL and *self-training* post-SFT. As shown in Table 5, both significantly enhance accuracy and efficiency. Without self-training, accuracy drops to 77.9% and latency increases to 17.6k, confirming that the 17k high-quality samples are vital for aligning teacher-curated data with Qwen3-8B's reasoning capabilities (Section 3).

Meanwhile, parallel rollout allows the policy to optimize execution efficiency directly, eliminating the mismatch between training rollouts and parallel inference. Note that self-training can be considered as a form of filtered behavior cloning or STaR (Zelikman et al., 2022b), and since our self-training data is always generated with parallel rollout in the ablations, extensive self-training can be considered as taking part in the role in reinforcement learning. When combined, these techniques enable ThreadWeaver to achieve 79.9% accuracy and a 16.9k token latency.

**Additional analysis.** We present a comprehensive analysis of ThreadWeaver's behavior in the Appendix, including per-stage ablations (Table 7) and efficiency metrics (Table 10). We also discuss the possibility of reward hacking (Appendix G) and provide a qualitative study in Appendix E.5 that illustrates ThreadWeaver spontaneously forking into distinct parallel strategies (e.g., algebraic vs. numerical) to solve complex problems, demonstrating adaptive task decomposition.

## 6 Related work

**Test-Time Scaling.** Test-time scaling has proven effective for enhancing LLM performance across various downstream tasks (Wei et al., 2022; Zelikman et al., 2022a; Gandhi et al., 2024; Snell et al., 2025; Team, 2025; Muennighoff et al., 2025). Prominent examples like OpenAI o1 (OpenAI, 2024)

and DeepSeek R1 (DeepSeek-AI, 2025) scale the length of reasoning trajectories via reinforcement learning. However, the sequential nature of autoregressive decoding means that longer outputs incur significant latency, which is unsustainable for complex domains. In contrast, parallelizing reasoning traces allows for adaptive allocation of computational budgets, achieving a superior Pareto frontier.

**Parallel Reasoning with Self-Consistency.** Parallel reasoning through self-consistency is a widely adopted technique to improve LLM inference robustness. By sampling multiple independent reasoning chains and aggregating the final answers, this approach provides a scaling dimension orthogonal to sequence length (Cobbe et al., 2021b; Wang et al., 2023; Chen et al., 2023). Related system results (e.g., Gemini Deep Think, Luong & Lockhart (2025)) also emphasize inference-time parallel compute scaling, though the exact mechanisms can differ from standard self-consistency. Variants refine this paradigm with confidence-guided aggregation and adaptive consensus (Fu et al., 2025; Wen et al., 2025). Nevertheless, because these threads are uncoordinated and independently executed, they often result in redundant computation and limited efficiency. Beyond voting-style aggregation, Madaan et al. (2025) propose Parallel-Distill-Refine (PDR), an externally orchestrated draft-distill-refine loop. ThreadWeaver instead learns branching and joining within a single reasoning trajectory.

**Structured (Agentic) Parallel Reasoning.** Structured parallel reasoning explores coordination via prompting-based agents or tree-of-thought frameworks (Yao et al., 2023; Du et al., 2023; Kim et al., 2024; Grand et al., 2025; Schroeder et al., 2024; Ning et al., 2024; Zhang et al., 2024; Hua et al., 2024; Zhuge et al., 2024; Teng et al., 2025). These methods decompose problems into subtasks using explicit structures like graphs or modular agents. While effective for symbolic or mathematical reasoning, they generally rely on hand-crafted heuristics and operate at inference time, lacking the scalability of learned reasoning strategies. Recent agentic formulations automate this process but still lack gradient-based optimization across reasoning branches (Wang et al., 2025; Rodionov et al., 2025; Chen et al., 2025).

**Adaptive Parallel Reasoning.** Recent training-based methods enable models to self-determine when to branch via special tokens. PASTA (Jin et al., 2025) and Multiverse (Yang et al., 2025b) utilize SFT and optionally preference optimization, but do not incorporate reinforcement learning (RL), which was shown critical for test-time scaling. APR (Pan et al., 2025) introduces an SFT+RL pipeline, but is restricted to synthesized datasets. Parallel-R1 (Zheng et al., 2025) incorporates RL; however, its SFT data is limited in terms of question complexity, resulting in final accuracy below that of official Qwen3 models (Yang et al., 2025a). Concurrent work NPR (Wu et al., 2025) also adopts RL but is

built upon instruct models rather than thinking models. As a result, both Parallel-R1 and NPR models have shorter CoTs compared to ThreadWeaver. In contrast, ThreadWeaver is built on thinking models (e.g., Qwen3-8B) and supports long CoT reasoning for higher accuracy. In addition, while Multiverse and NPR require non-trivial modifications to the underlying inference engine, ThreadWeaver is compatible with standard sequential engines for easy deployment.

## 7 Conclusion

We present ThreadWeaver, an adaptive parallel reasoning framework that addresses key limitations of prior approaches through three core contributions: 1) a two-stage parallel trajectory generator that enables scalable fine-tuning data construction, 2) a trie-based training–inference co-design that supports deployment on standard LLM inference engines, and 3) P-GRPO, an RL framework that jointly optimizes reasoning accuracy and inference acceleration. Across six math reasoning benchmarks, ThreadWeaver achieves up to a $1.53\times$ token-level latency reduction relative to sequential baselines while achieving accuracy comparable to cutting-edge thinking models, establishing a new Pareto frontier between efficiency and reasoning performance.

## Acknowledgements

We thank Jiayi Pan and Abhimanyu Dubey for helpful early discussions. We thank Xinyu Yang and Beidi Chen for answering our questions about the Multiverse paper and codebase. We thank Robert Nishihara, Richard Liaw, Jiajun Yao, and the Anyscale team for providing us with an early implementation of Ray with IPv6 support, even though we ultimately adopted a different approach due to timing constraints. We thank Vignesh Ramanathan and Natalia Neverova for their support of this project.

## Impact Statement

This paper presents work whose goal is to advance the field of Machine Learning. There are many potential societal consequences of our work, none of which we feel must be specifically highlighted here.

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

# A  Preliminaries

## A.1  Adaptive Parallel Reasoning

Many reasoning problems naturally contain parallelizable structure, such as independent sub-cases, alternative solution paths, or self-reflection on partial results that can be handled along with exploration. Adaptive parallel reasoning (APR) (Pan et al., 2025) is a reasoning paradigm in which a language model dynamically decides when to exploit such structure within a single inference trajectory.

Formally, let $\pi_\theta$ be a policy over an extended action space that includes text token emissions and two control actions, spawn and join. Executing spawn at time $t$ creates $m \geq 1$ child threads $\{\tau^{(j)}\}_{j=1}^m$ that evolve independently with no cross-thread reads under $\pi_\theta$ until a join. At join, a designated reducer aggregates the child outputs (e.g., via concatenation or a learned summarization pattern) and returns the resulting information back to the parent thread, which then continues decoding sequentially.

APR (Pan et al., 2025; Yang et al., 2025b; Zheng et al., 2025) differs from *inter-trajectory* parallelism such as self-consistency and majority vote (Wang et al., 2023), where multiple complete chains are sampled independently and combined only at the end. In APR, branching and joining occur *inside* a single trajectory with explicit coordination across threads. This allows the model to parallelize decomposable subproblems while maintaining a single coherent solution path. Our framework, ThreadWeaver, follows this paradigm.

## A.2  Test-time Speed-Accuracy Pareto Frontier

Test-time scaling methods trade additional inference compute for higher accuracy. Prior evaluations typically emphasize accuracy alone, e.g., by generating longer chains of thought or sampling more trajectories for majority vote, while overlooking the accompanying latency costs.

We instead adopt a **speed-accuracy Pareto view**: given a difficult problem and additional compute, we ask whether a method can *reduce the end-to-end time from receiving a user prompt to producing the final answer*, rather than merely increasing accuracy at the cost of slower inference. In our setting, latency is proxied by **token latency**, defined as the number of tokens on the longest thread, or critical path, of a parallel inference trajectory (Section 5.1). For sequential models, this reduces to the total number of generated tokens. This metric correlates well with wall-clock time when parallelism is utilized at inference. Unlike Multiverse (Yang et al., 2025b) and Parallel-R1 (Zheng et al., 2025), which focus on measuring latency per token and the degree of parallelism within a parallel trajectory, we emphasize the speedup to answer generation by comparing the token latency of a parallel model with a sequential model with comparable reasoning capability under the same prompts.

Viewing existing approaches under this measurement reveals sharp differences. Self-consistency (Wang et al., 2023) and other ensemble approaches improve robustness by sampling multiple full solutions in parallel, yet the critical-path latency remains that of an entire trajectory. Moreover, because trajectories are sampled independently without coordination, they often replicate large portions of the same reasoning process, leading to substantial redundant computation. In contrast, APR introduces structured *intra-run* branching: the model decomposes a problem into multiple threads that execute concurrently and then rejoin. When this decomposition aligns with the underlying problem structure, the longest thread can be significantly shorter than a monolithic long chain-of-thought, yielding strictly lower token latency.

## A.3  GRPO Preliminary

Group Relative Policy Optimization (GRPO) is a policy-gradient method that computes advantages with intra-prompt normalization (Shao et al., 2024). For a prompt $p$, let $\mathcal{G}_p = \{r_1, \ldots, r_k\}$ be the scalar rewards for $k$ trajectories sampled from $p$. The group-normalized advantage for trajectory $i$ is given in Equation (1), which yields advantages that are comparable across prompts, mitigating reward-scale and difficulty variance.

While GRPO can be paired with a PPO-style clipping surrogate (Schulman et al., 2017), in our setting we use fully *on-policy* updates (single rollout, single optimization step), so the importance ratio is effectively 1 and clipping is inactive. We thus optimize the score-function objective

$$\mathcal{J}_{\text{GRPO}}(\theta) \;=\; \mathbb{E}_{p,i,\tau_p^{(i)} \sim \pi_\theta}\Big[ A_{p,i}^{\text{GRPO}} \sum_{t=1}^{T_{p,i}} \log \pi_\theta(a_t \mid s_t) \Big], \tag{8}$$

whose gradient is obtained from the standard REINFORCE estimator. In practice, we minimize the pseudo-loss

$$\mathcal{L}_{\text{GRPO}}(\theta) \;=\; -\frac{1}{\sum_{p\in\mathcal{B}}\sum_{i=1}^{k} T_{p,i}} \sum_{p\in\mathcal{B}}\sum_{i=1}^{k} A_{p,i}^{\text{GRPO}} \sum_{t=1}^{T_{p,i}} \log \pi_\theta(a_t \mid s_t), \tag{9}$$

where $T_{p,i}$ is the number of completion tokens for the trajectory $\tau_p^{(i)}$. This loss has gradient $-\nabla_\theta \mathcal{J}_{\text{GRPO}}$. Note that we follow DAPO (Yu et al., 2025) and normalize over tokens across sequences in the batch ($\sum_{p\in\mathcal{B}}\sum_{i=1}^{k} T_{p,i}$).

## B   Derivations for P-GRPO

### B.1   From clipped GRPO to the score-function objective

For completeness, we first establish that P-GRPO with strictly on-policy sampling reduces to a score-function (REINFORCE) estimator with a baseline. Define the PPO-style clipped GRPO surrogate over a batch $\mathcal{B}$ with group size $k$:

$$\mathcal{J}_{\text{clip}}(\theta) \;=\; \frac{1}{|\mathcal{B}|}\sum_{p\in\mathcal{B}}\frac{1}{k}\sum_{i=1}^{k} \mathbb{E}_{\tau_p^{(i)}\sim\pi_{\theta_{\text{old}}}}\Big[\min\big(r_\theta(\tau_p^{(i)})A_{p,i},\; \text{clip}(r_\theta(\tau_p^{(i)}),1-\epsilon,1+\epsilon)\,A_{p,i}\big)\Big], \tag{10}$$

where $A_{p,i}$ is the advantage computed via GRPO and the importance ratio is:

$$r_\theta(\tau) \;=\; \frac{\pi_\theta(\tau)}{\pi_{\theta_{\text{old}}}(\tau)}. \tag{11}$$

The gradient of the unclipped objective with respect to $\theta$ is:

$$\nabla_\theta \mathbb{E}\big[r_\theta(\tau)A\big] \;=\; \mathbb{E}\big[A\,\nabla_\theta r_\theta(\tau)\big] \;=\; \mathbb{E}\big[A\,r_\theta(\tau)\,\nabla_\theta \log \pi_\theta(\tau)\big]. \tag{12}$$

In our setting, we perform a single gradient update immediately after data collection (fully on-policy), implying $\theta_{\text{old}} = \theta$. Consequently, $r_\theta(\tau) = 1$ identically, rendering the clipping operation inactive. Substituting $r_\theta(\tau) = 1$ into Equation (12) yields the standard score-function gradient:

$$\nabla_\theta \mathcal{J}_{\text{P−GRPO}}(\theta) \;=\; \mathbb{E}_{\tau\sim\pi_\theta}\Big[A(\tau)\nabla_\theta \log \pi_\theta(\tau)\Big]. \tag{13}$$

### B.2   Factorization of Parallel Reasoning Trajectories

To implement Equation (13) efficiently for tree-structured parallel reasoning, we factorize the probability of the trajectory. Let a parallel trajectory $\tau_p^{(i)}$ be represented as a sequence of $M_i$ disjoint generation segments (or units) produced by the state machine:

$$\tau_p^{(i)} = \big((\text{cont}^{(i,1)}, \text{comp}^{(i,1)}), \dots, (\text{cont}^{(i,M_i)}, \text{comp}^{(i,M_i)})\big)$$

Here, each unit $m$ consists of a context $\text{cont}^{(i,m)}$ (which may include the prompt and previously generated segments) and a newly generated completion $\text{comp}^{(i,m)}$. Without loss of generality, in this derivation, we ignore tokens that we do not apply loss on (e.g., the user prompt and the tokens inserted by the state machine orchestrator). Under this assumption, the set of all completion segments $\{\text{comp}^{(i,m)}\}_{m=1}^{M_i}$ forms a partition of all tokens generated by the model in the trajectory $\tau_p^{(i)}$.

By the chain rule of probability, the log-probability of the entire trajectory is the sum of the log-probabilities of its generated tokens. Since each completion segment $\text{comp}^{(i,m)}$ is generated conditioned on its specific history $\text{cont}^{(i,m)}$, we have:

$$\log \pi_\theta(\tau_p^{(i)}) \;=\; \sum_{m=1}^{M_i} \log \pi_\theta\big(\text{comp}^{(i,m)} \mid \text{cont}^{(i,m)}\big). \tag{14}$$

This factorization holds regardless of the parallel structure. For example, a shared `<Outline>` segment is simply the completion of the first unit (conditioned on the prompt), while subsequent `<Thread>` segments are completions of later units (conditioned on the prompt and the Outline). The independence of parallel threads is naturally captured by the fact that their contexts branch from the same parent node in the trie.

## B.3 Broadcasted Credit Assignment

We now show that broadcasting the trajectory-level advantage to all tokens in all threads is mathematically justified by relating it to the gradient of the total trajectory loss. Substituting the factorization from Equation (14) into the gradient objective in Equation (13):

$$
\begin{aligned}
\nabla_\theta &\mathcal{J}_{\text{P-GRPO}}(\theta) \\
&= \mathbb{E}_{\tau_p^{(i)} \sim \pi_\theta} \left[ A_{p,i}^{\text{P-GRPO}} \nabla_\theta \log \pi_\theta(\tau_p^{(i)}) \right] \\
&= \mathbb{E}_{\tau_p^{(i)} \sim \pi_\theta} \left[ A_{p,i}^{\text{P-GRPO}} \nabla_\theta \sum_{m=1}^{M_i} \log \pi_\theta\big(c_p^{(i,m)} \mid x_p^{(i,m)}\big) \right] \\
&= \mathbb{E}_{\tau_p^{(i)} \sim \pi_\theta} \left[ \sum_{m=1}^{M_i} A_{p,i}^{\text{P-GRPO}} \nabla_\theta \log \pi_\theta\big(c_p^{(i,m)} \mid x_p^{(i,m)}\big) \right],
\end{aligned}
\tag{15}
$$

where $c_p^{(i,m)} = \text{comp}^{(i,m)}$ and $x_p^{(i,m)} = \text{cont}^{(i,m)}$ for brevity. Equation (15) demonstrates that the gradient of the full trajectory objective is simply the sum of the gradients for each completion segment, weighted by the *same* scalar trajectory advantage $A_{p,i}^{\text{P-GRPO}}$.

In our implementation, we approximate the expectation using a Monte Carlo estimate over the batch $\mathcal{B}$. The broadcasting operation described in Section 4.1 assigns the scalar $A_{p,i}^{\text{P-GRPO}}$ to every token in every completion $\text{comp}^{(i,m)}$. When we compute the loss on these tokens weighted by $A_{p,i}^{\text{P-GRPO}}$, we are computing exactly the inner term of the summation in Equation (15). Thus, P-GRPO's advantage broadcasting is not a heuristic approximation. It is the policy gradient estimator for the parallel trajectory reward structure.

## C  Implementation Details

We provide implementation details for ThreadWeaver training pipeline.

**Data Preparation.** Training prompts are sourced from the Polaris 53k dataset (An et al., 2025), which contains 53,291 high-quality math reasoning problems. Each prompt contains a question and a reference answer. We perform initial inference with Qwen3-8B to obtain a single reasoning trajectory per prompt. In all generated trajectories, 36,296 (68.1%) contain both a correct final answer (validated by symbolic evaluation) and a valid reasoning structure. These form the base candidate trajectory pool for SFT.

**Trie-based Sequence Merging** In addition to the single branching case in the example provided by Figure 4, our trie-based sequence merging also works on trajectories that contain multiple `<Parallel>` blocks interleaved with sequential reasoning segments. We explain how trie-based sequence merging works with multiple `<Parallel>` blocks in detail.

As illustrated in Figure 6, we follow the same three steps as in the single-block case: extract all ⟨context, completion⟩ units that the inference state machine would issue, insert their token sequences into a shared prefix tree, and traverse the trie to obtain a single packed sequence with an ancestor-only attention mask. Loss is still applied only to completion tokens, so the total number of supervised tokens matches training on each unit independently. Because the model also supports hybrid reasoning (i.e., it supports pure autoregressive decoding by skipping the state machine), we additionally treat the full trajectory as a single ⟨context, completion⟩ unit, with the user prompt as the context and the full model completion as the completion. For simplicity, we omit this full trajectory unit in Figure 4 and Figure 6, as including the unit is only to allow the model to perform sequential reasoning with standard autoregressive decoding.

**Detailed State-Machine Specification.** The inference orchestrator implements a state machine with the following detailed phases, starting with a sequential phase with the user prompt wrapped in a chat template defined by the model as the initial context:

1. **Sequential phase.** Decode sequentially until `</Outlines>` is generated, which is registered as a stop token for the sequential phase.
2. **Parse outlines.** Extract the numbered `<Outline>` entries from the preceding `<Outlines>` block.
3. **Parallel phase.** For each `<Outline>` entry $i$, issue a completion request with `<Thread>` **i:** appended to the previous context that includes all tokens that the LLM generated previously, including the `<Outlines>` block. Generate

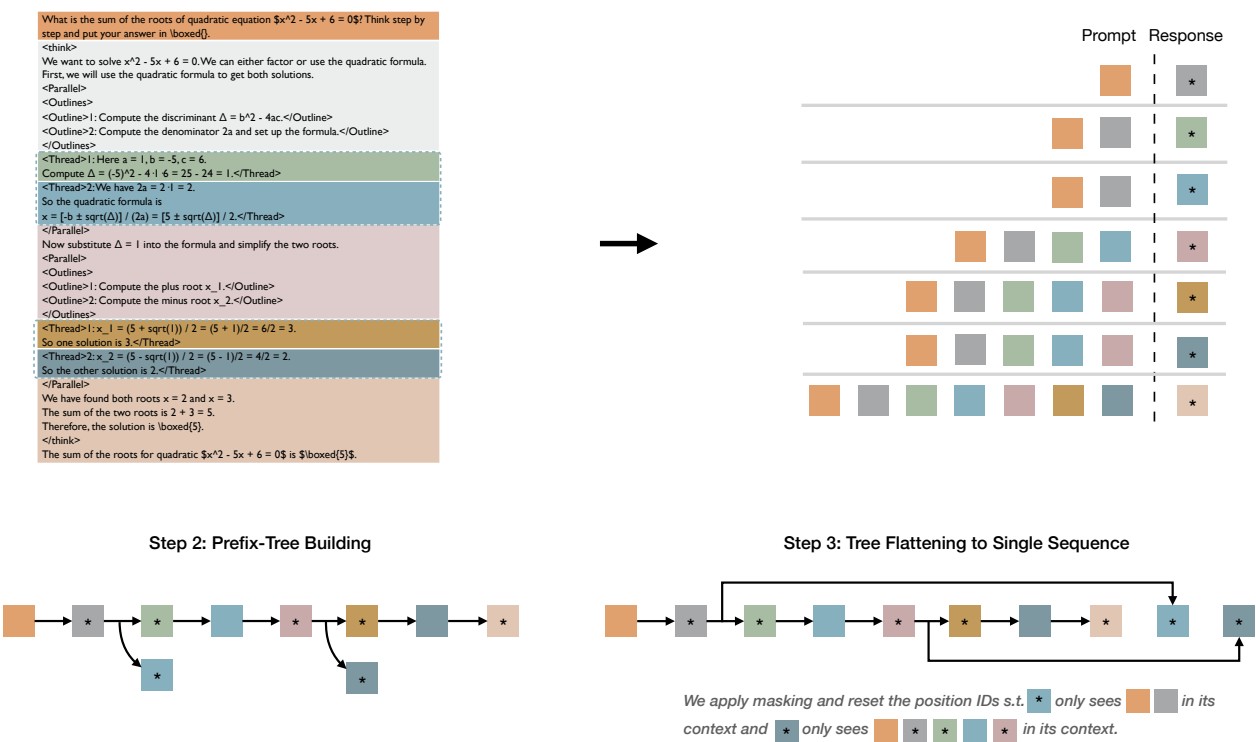

*Figure 6.* Trie-based sequence merging with more than one parallel block. We still build a prefix-tree (trie) based on the <context, completion> pairs from the training example and then flatten the tree into one packed training sequence with an ancestor-only attention mask.

completions independently until `</Thread>`, which is registered as a stop token for independent generation in the parallel phase. Threads do not attend to each other as they are treated as different requests by the inference engine.

4. **Join.** Concatenate the context (prompt, outlines, and all thread texts) and append `</Parallel>` to form the context for the next round of text completion. While we expose all the reasoning in subthreads to the main thread, further optimizations are possible to only return necessary information for subsequent computation, reducing the context usage for reasoning.

5. **Continue.** Proceed sequentially until the next `</Outlines>` token or the `[EOS]` token.

Because we introduce no changes to the LLM architecture or context representation, our method inherits existing serving optimizations (e.g., paged attention, prefix caching) and runs out of the box on common LLM inference stacks that expose a text-completion API. Note that common inference engines such as vLLM and SGLang support prefix caching, which prevents regenerating the KV cache (i.e., rerunning prefilling) for context tokens if they share prefix with a prior request. When the host API supports taking and returning token IDs, all steps above can operate directly on token IDs without text decoding for additional efficiency. The main potential sources of overhead are repeated API calls, which can increase queueing and scheduling costs especially in trajectories with many short branches, and needing to prefill for threads beyond the first one, which can be expensive when later branches are much longer. In our experiments, these factors did not become bottlenecks, as reflected in the wall-clock speedups reported in Section 5.5, in part because we use an efficient vLLM-based implementation with low scheduling overhead and because prefilling is significantly cheaper than decoding.

**Supervised Fine-tuning (SFT).** From the 36,296 valid trajectories, we sample 1,000 for rewriting using GPT-5 (OpenAI, 2025). Each sample is rewritten with a structured instruction prompt to improve readability and ensure canonical parallelization, while preserving the exact step order and final boxed answer. After automatic filtering for format and semantic alignment, we use the remaining 959 high-quality rewritten trajectories as the training data for SFT.

SFT is performed for 8 epochs with the following hyperparameters:

*Table 6.* Statistics for the three training trajectory datasets. "Sequential" are raw Qwen3–8B trajectories subsampled from the 36k trajectories with the correct answers; "First-stage" are GPT–5–rewritten parallel traces; "Self-train" are trajectories generated by the model trained with data in the first stage and filtered as in Section 3. Parallel and acceleration ratios are computed per trajectory then averaged.

|  | Sequential Trajectories (sampled) | First-stage | Self-train |
|---|---|---|---|
| Size (format-correct) | 1000 | 959 | 17,491 |
| Avg. parallel ratio | 0.00 | 0.77 | 0.68 |
| Avg. longest seq. length | 10,377 | 8,278 | 9,153 |
| Avg. total tokens | 10,377 | 11,143 | 11,176 |
| Self-parallelism speedup | 1.00 | 1.35 | 1.22 |
| Avg. # parallel blocks | 0.0 | 6.26 | 4.00 |
| Avg. parallel path length | – | 541 | 969 |
| Avg. tokens per parallel block | – | 1,542 | 2,458 |
| Avg. threads per parallel block | – | 2.58 | 2.46 |

- **Optimizer:** AdamW with $\beta_1 = 0.9$, $\beta_2 = 0.95$, and weight decay 0.01.
- **Learning rate:** initial $1 \times 10^{-5}$ with 5% warmup and cosine decay.
- **Batch size:** 16.
- **Sequence length:** maximum 40,960 tokens.
- **Loss:** standard cross-entropy on the tokens with loss mask specified in Section 2.

**Self-training.** We apply the SFT-trained model to the full 53k Polaris prompt set to generate new reasoning trajectories. Each trajectory is automatically scored by a dual-criteria filter:

1. **Answer correctness:** whether the final boxed expression matches the ground truth, using symbolic equivalence checks implemented via SymPy.
2. **Structural validity:** whether all `<Parallel>` and `</Parallel>` markers form a valid structure, ensuring no partial threads or misuse of special tokens.

The trajectories generated with this process are used for the second-stage fine-tuning.

The self-training SFT stage uses identical hyperparameters to the initial SFT, except we only run for 1 epoch.

**Reinforcement Learning.** The final stage employs the proposed **Parallel-GRPO (P-GRPO)** algorithm. Parallel rollouts are performed so that each of the prompts is sampled 8 times in each batch, and sequence-level advantages are normalized by group mean within each prompt. Additional hyperparameters are listed as follows:

- **Optimizer:** Adam with learning rate $5 \times 10^{-6}$, no weight decay.
- **Batch size:** 128 (8 rollouts $\times$ 16 prompts).
- **KL penalty:** No KL penalty is applied.
- **Temperature:** 1.0 during rollout and at evaluation.
- **Precision:** bf16.

We run the training for 350 optimization steps. We use vLLM (Kwon et al., 2023) as our parallel inference backend. We leverage fully-sharded data parallel (FSDP, Zhao et al. (2023)) in distributed training.

**Evaluation Setup.** Evaluation is conducted across six datasets: AIME24 (Mathematical Association of America, 2024), AIME25 (Mathematical Association of America, 2025), AMC23 (Mathematical Association of America, 2023), MATH500 (Hendrycks et al., 2021), Minerva Math (Lewkowycz et al., 2022), and OlympiadBench (He et al., 2024). Inference is conducted with total number of tokens set to 40k across the parallel branches.

## D  Dataset Statistics

We report core statistics of the three trajectory datasets used in ThreadWeaver training: **1)** the original sequential Qwen3–8B trajectories on Polaris–53K, **2)** the dataset with parallel annotations processed by GPT-5 (OpenAI, 2025) and used in the first stage of training (Section 3), and **3)** the larger self–training dataset generated by the model itself (Section 3).

The original trajectories are long and purely sequential, with no acceleration from parallelization. Rewriting with GPT-5

*Table 7.* Stagewise performance on AIME 24. *Format correctness* checks proper closure of all opened `Parallel` blocks and well-formed thread markers (a proxy for structural stability). Common reasons for format error are truncation under token budgets and early stopping without closing the parallel block. Correctness of the final answer depends only on the last `\boxed{}`.

| Model | Format Correctness | Accuracy (Avg@32) |
|---|---|---|
| Qwen3-8B (Baseline without SFT) | – | 76.0% |
| Qwen3-8B + 1st SFT (959 SFT examples) | 56.4% | 74.5% |
| Qwen3-8B + 1st SFT + RL | 41.9% | 77.9% |
| Qwen3-8B + self-training (17k SFT examples) | 77.0% | 74.0% |
| Qwen3-8B + self-training + RL | 72.4% | **79.9%** |

introduces substantial parallel structure: about 77% of tokens lie in parallel blocks, and the critical path is on average about 26% shorter than the total token count, despite a slight increase in overall length. This corresponds to an idealized self-parallelism speedup of roughly $1.35\times$ (computed by dividing the token latency with total number of tokens averaged across the dataset) and provides strong supervision for the fork-join format. We also observe rich structural annotations, with about 2.6 threads per parallel block on average.

As discussed in Section 3, however, the rewritten traces are produced post hoc from complete solutions and do not always match how the model naturally decomposes problems at inference time: the strong teacher can rely on global hindsight and sometimes parallelizes by skipping or compressing intermediate steps that the student needs. To close this distribution gap, we perform self-training with the trajectories extracted by the model obtained from the first stage under our proposed parallel inference state machine, then filter trajectories to keep only structurally valid and answer-correct samples, with length and acceleration constraints. The resulting self-training corpus is much larger (17,491 trajectories). Since self-training only takes parallel trajectories that the smaller student model is able to generate ad hoc during reasoning, where it does not have full view of the overall trajectories except for the tokens it has generated, self-training reduces the self-parallelism speedup to $1.22\times$ while remaining strongly parallel, and better reflects what the model itself can reliably parallelize.

Together, our dataset from the two-stage data pipeline provides robust training signal for both supervised fine-tuning and prepares the model for the subsequent reinforcement learning.

# E    Additional Results

## E.1    Performance Across Training Stages

We evaluate performance on AIME 24 across successive training stages, measuring both *accuracy* (Avg@32) and *format correctness*, which reflects whether all opened `Parallel` blocks are properly closed. Format correctness serves as a proxy for structural stability, though it does not directly affect answer accuracy since the final result depends only on the last `\boxed{}` expression.

As shown in Table 7, accuracy drops after the first SFT on 959 examples (74.5%), with limited format correctness (56.4%). This indicates that small-scale supervised data is insufficient for mastering the parallel reasoning. Expanding to 17k examples via self-training substantially improves format correctness (77.0%) while maintaining similar accuracy, showing that model-generated data helps the model internalize structural consistency. Reinforcement learning (RL) further improves accuracy in both cases (77.9% and 79.9%), but format correctness decreases slightly when initialized from the smaller dataset. This is expected since RL directly optimizes outcome and efficiency rather than explicit structural closure. Overall, self-training yields a more stable structural foundation, enabling RL with parallel rollout to achieve higher accuracy and lower token latency.

## E.2    Cold-start with Sequential Data Mixture

Multiverse (Yang et al., 2025b) proposed to cold-start with a mixture of sequential and parallel data in order to reduce the degradation that can occur when directly fine-tuning on parallel reasoning trajectories. We also ablated such cold-start strategies in our SFT stage, using a dynamic mixture between two datasets: a purely sequential dataset $\mathcal{D}_{seq}$ built from autoregressive Qwen3-8B trajectories, and a parallel dataset $\mathcal{D}_{par}$ built from our parallelized trajectories rewritten from $\mathcal{D}_{seq}$.

*Table 8.* Ablation of sequential–parallel cold-start mixing strategies during SFT on AIME 24. "No sequential mixture cold start" uses only parallel data in SFT. "Multiverse-style mixing recipe" follows a non-gradual schedule that switches between mostly sequential, mixed, and mostly parallel phases. "Linear mixing recipe" uses an eight-stage curriculum that linearly increases the proportion of parallel data across epochs. Neither strategy improves over the parallel-only SFT setting, suggesting that additional sequential data does not help Qwen3-8B transition to parallel reasoning once it already has strong sequential capabilities.

| Model | Accuracy (Avg@32) |
|---|---|
| Qwen3-8B (No sequential mixture cold start) | **74.5%** |
| Qwen3-8B (Cold start with Multiverse-style mixing recipe) | 73.4% |
| Qwen3-8B (Cold start with linear mixing recipe) | 73.1% |

We then constructed curriculum schedules over SFT epochs that control how often each dataset is sampled. Concretely, we considered two families of curricula: a *staged* schedule that switches in a few coarse phases, following (Yang et al., 2025b), and a gradual schedule that increases the share of parallel data linearly over eight epochs. Both variants are trained for 8 epochs, following the training schedule of ThreadWeaver, for fair comparison. Only SFT is conducted for this comparison to isolate the effect of each cold-start strategy.

**Staged mixture (Multiverse-style mixing recipe).** For the staged schedule, we follow (Yang et al., 2025b) and organize SFT into three phases:

- **Phase 1: fully sequential phase.** The model is trained for 2 epochs using only $\mathcal{D}_{\text{seq}}$.
- **Phase 2: mixed phase.** The model is trained for 4 epochs on mixtures of sequential and parallel data, implemented as four one-epoch subphases with different fixed mixing ratios.
- **Phase 3: fully parallel phase.** The model is trained for 2 epochs using only $\mathcal{D}_{\text{par}}$.

Conceptually, the curriculum can be summarized as

$$\underbrace{\mathcal{D}_{\text{seq}}}_{\text{fully sequential phase}} \rightarrow \underbrace{\alpha \, \mathcal{D}_{\text{seq}} + (1 - \alpha) \, \mathcal{D}_{\text{par}}}_{\text{mixed phase}} \rightarrow \underbrace{\mathcal{D}_{\text{par}}}_{\text{fully parallel phase}} \,,$$

where $\alpha \in (0, 1)$ controls the fraction of sequential data. In Multiverse's implementation, $\alpha$ is not a single constant but is implicitly determined by how the two datasets are sliced and concatenated. The four mixed subphases correspond to:

- A first mixed epoch that uses $90\%$ $\mathcal{D}_{\text{seq}}$ and $10\%$ $\mathcal{D}_{\text{par}}$ (effectively $\alpha = 0.9$)
- Two middle mixed epochs that each combine $50\%$ $\mathcal{D}_{\text{seq}}$ and $50\%$ $\mathcal{D}_{\text{par}}$ (effectively $\alpha = 0.5$)
- A final mixed epoch that uses $10\%$ $\mathcal{D}_{\text{seq}}$ and $90\%$ $\mathcal{D}_{\text{par}}$ (effectively $\alpha = 0.1$)

Intuitively, Phase 2 therefore sweeps from mostly sequential, through approximately balanced mixtures, to mostly parallel before transitioning to the fully parallel Phase 3.

**Gradual mixture (linear mixing recipe).** In the second variant, we construct a smooth curriculum over eight stages, each corresponding to roughly one epoch. We again treat $\mathcal{D}_{\text{seq}}$ as the compatible supervised signal and $\mathcal{D}_{\text{par}}$ as the target format. At epoch $k \in \{1, \ldots, 8\}$ we set a target fraction for parallel data

$$\lambda_k = 0.125 \times k,$$

and sample a batch that contains a fraction $\lambda_k$ from $\mathcal{D}_{\text{par}}$ and $(1 - \lambda_k)$ from $\mathcal{D}_{\text{seq}}$. This produces the progression

$$12.5\% \text{ parallel} \rightarrow 25\% \rightarrow 37.5\% \rightarrow 50\% \rightarrow 62.5\% \rightarrow 75\% \rightarrow 87.5\% \rightarrow 100\% \text{ parallel},$$

so the model starts in a regime that is almost fully sequential and ends in a purely parallel regime, with the proportion of parallel examples changing smoothly at each stage. Within each stage we shuffle both datasets and sample without additional bias, so the only variable that changes across epochs is the global ratio between sequential and parallel trajectories.

In both settings, the goal is to cushion the transition from sequential to parallel reasoning, either by a small number of coarse phases (staged mixture) or by a fine-grained eight-epoch ramp (gradual mixture). However, as shown in Table 8, neither curriculum yielded clear benefits over directly training on parallel data without any sequential mixture. We hypothesize that this is because Qwen3-8B is already a strong reasoning model with robust sequential behavior. When sequential data is continually mixed in, the model receives a strong training signal that reinforces its existing autoregressive reasoning style,

*Table 9.* Comparison of base model performance on AIME 24 (Avg@32). AR SFT uses sequential Qwen3-8B trajectories; Our SFT uses GPT-5-rewritten parallel trajectories.

| Model | Accuracy (Avg@32) |
|---|---|
| Qwen3-4B-Base + AR SFT (1k SFT examples) | 37.4% |
| Qwen3-4B-Base + Our first-stage SFT (959 SFT examples) | 32.4% |
| Qwen3-8B-Base + AR SFT (1k SFT examples) | 55.1% |
| Qwen3-8B-Base + Our first-stage SFT (959 SFT examples) | 45.5% |

which slows down the adaptation to the new parallel format. In other words, the mixed curricula make it harder for the model to fully commit to parallel structure, so the additional sequential examples do not translate into better performance in our setting, despite their success as a cold-start recipe in Multiverse.

### E.3 ThreadWeaver Reuses Existing Reasoning Capabilities for Parallel Reasoning

To verify that ThreadWeaver can reuse strong reasoning capabilities from the base model without degradation, we compare Qwen3-4B-Base and Qwen3-8B-Base trained on our SFT data from the first stage (959 GPT-5 rewritten trajectories) against the same base models trained on standard autoregressive (AR) SFT data (randomly sampled 1k correct sequential Qwen3-8B trajectories). Both variants are trained for 8 epochs. As shown in Table 9, the gap between AR SFT and our SFT is roughly 5% for both 4B and 8B models. This demonstrates that ThreadWeaver preserves most of the base model's reasoning ability despite structural changes and remains compatible with standard reasoning models without extensive re-training. Performance gains from the stronger 8B base carry over to the parallel setting, confirming that our approach scales with model capacity.

### E.4 Detailed Experimental Analysis

#### E.4.1 Critical-Path Speedup Against Sequential Baseline as Robust Efficiency Measurement

We find that metrics that only quantify the amount of parallelization within a parallel reasoning trajectory do not accurately reflect true inference-time acceleration obtained when switching from sequential to parallel reasoning. To rigorously assess efficiency gains, we recommend making comparisons directly against *sequential baselines* trained and evaluated under comparable settings.

As summarized in Table 10, we report two quantities: 1) *speedup vs. sequential baseline* ($L_{\text{baseline longest}}/L_{\text{ours longest}}$), and 2) *self-parallelism speedup* ($L_{\text{ours total}}/L_{\text{ours longest}}$). Note that for the sequential baseline, the token latency is the same as the total number of generated tokens.

In three of six benchmarks, the self-parallelism speedup significantly exceeds the speedup measured against sequential baselines. This contrast is informative: AIME 25 shows a clear self-parallelism gain ($1.18\times$) but only a modest $1.03\times$ improvement over the baseline.

Thus, self-parallelism speedup is not a complete efficiency metric. For deployment, the decisive measure is the *critical-path speedup vs. the sequential baseline*. As in other forms of parallelization, APR introduces overhead, including the need to generate outline tokens and additional redundancy because threads operate without visibility into each other. This motivates benchmarking ThreadWeaver against sequential RL on Qwen3-8B.

#### E.4.2 High-Quality SFT Data is Critical for Model Performance

Both our SFT dataset and the Multiverse dataset are constructed from LLM-generated trajectories, but differ in data origin and model alignment. While Multiverse (Yang et al., 2025b) uses trajectories from s1 (Muennighoff et al., 2025) generated by DeepSeek-R1 (DeepSeek-AI, 2025), our dataset is generated directly from Qwen3-8B outputs and then lightly rewritten by GPT-5 (OpenAI, 2025). To compare these sources fairly, we fine-tune Qwen3-8B on each dataset under identical settings (about 1k samples, 8 epochs). In this ablation, we expand the internal nested parallelization structures in the Multiverse dataset to match the behavior of our model, following how Yang et al. (2025b) curates data for autoregressive baselines. This is not expected to impact accuracy significantly since the accuracy only fluctuates slightly even when all parallelization structures are expanded into sequential structures, as reported in Yang et al. (2025b). To further ablate the impact of

*Table 10.* **Two notions of efficiency.** Speedup vs. sequential baseline reflects switching from sequential to parallel reasoning. Self-parallelism speedup reflects internal parallel breadth. Speedups are computed per-problem and averaged.

| Dataset | ThreadWeaver Longest | Seq. Baseline Longest | Speedup vs. Seq. Baseline | ThreadWeaver Total Tokens | Self-Parallelism Speedup |
|---|---|---|---|---|---|
| AIME24 | 16.9k | 19.4k | 1.14× | 21.1k | **1.25×** |
| AIME25 | 24.0k | 24.6k | 1.03× | 27.3k | **1.18×** |
| AMC23 | 12.0k | 13.8k | 1.16× | 15.0k | **1.25×** |
| MATH500 | 6.4k | 7.2k | **1.23×** | 7.7k | 1.22× |
| Minerva Math | 7.3k | 10.6k | **1.53×** | 8.4k | 1.14× |
| OlympiadBench | 12.8k | 15.2k | 1.21× | 15.5k | **1.22×** |

*Table 11.* SFT data comparison on AIME 24. Our data yields stronger performance. *Results from (Zheng et al., 2025) (Avg@16); others use Avg@32.

| Model | Acc. |
|---|---|
| Qwen3-8B + Multiverse SFT (1k samples) | 62.2% |
| Qwen3-8B + Our SFT (959 samples) | **74.5%** |
| Qwen3-4B-Base + Parallel-GSM8K (7.5k) | 10.6%* |
| Qwen3-4B-Base + Parallel-GSM8K + RL | 19.4%* |
| Qwen3-4B-Base + Our SFT (959 samples) | **32.4%** |

data quality, we also fine-tune Qwen3-4B-Base, the backbone used in Parallel-R1 (Zheng et al., 2025), on our dataset. Parallel-R1 uses its proposed Parallel-GSM8K, curated with DeepSeek-R1-0528-Qwen3-8B (DeepSeek-AI, 2025) based on GSM8K (Cobbe et al., 2021a).

As shown in Table 11, training on the Multiverse dataset yields lower performance despite its stronger teacher model. This indicates that dataset-model compatibility is more important than teacher strength in our setting. We hypothesize that DeepSeek-R1 trajectories differ stylistically from Qwen3-8B's native reasoning patterns, introducing distributional mismatch and limiting generalization. In contrast, our dataset preserves Qwen3-8B's reasoning semantics while primarily adjusting structural organization to expose parallel reasoning patterns. This allows the model to acquire parallel structure effectively without disrupting its underlying reasoning behavior.

Our SFT model also substantially outperforms Parallel-R1. This is partially due to the fact that Parallel-GSM8k is curated from grade school math dataset GSM8k (Cobbe et al., 2021a), which is likely too simple compared to competition-level math benchmarks. Although Parallel-R1 applies RL, the gains are modest and performance remains much lower. This suggests that *RL cannot fully compensate for lower-quality SFT data*. Both comparisons indicate that high-quality, model-aligned SFT data is a key factor in downstream performance and parallelization ability, whereas additional RL provides only limited gains when the SFT distribution is poorly matched to the target data distribution.

### E.5  Qualitative Study

To illustrate how ThreadWeaver works in practice, we analyze a trigonometric evaluation example from MATH500 below, in which the model spontaneously forks into multiple reasoning branches to explore distinct solution strategies and joins the reasoning threads to continue the next steps of reasoning. Due to the excessive length of the reasoning trajectories, we use brackets with summarized content in place of bulk reasoning text.

In the first parallelization block, the model explores two distinct approaches to the problem. While the second one reaches a numerical solution, the first one ends up switching to another direction that is more promising. Then in the second parallelization block, the first branch continues to explore the direction that the first branch of the first parallelization block is exploring to get to an exact answer, while the second branch performs self-reflection to double check the numerical answer. These branches parallelize well because they operate on disjoint subgoals that are independent of each other given the shared context. Finally, the main thread integrates the information from all the previous reasoning results and output solutions to the user. This example highlights how ThreadWeaver enables adaptively parallelized exploration and self-reflection within a single inference trajectory.

---

ThreadWeaver Example: Trigonometric Expression Evaluation

**Prompt:**
Evaluate
$$\sin(\arcsin 0.4 + \arcsin 0.5) \cdot \sin(\arcsin 0.5 - \arcsin 0.4).$$

Let's think step by step and output the final answer within \boxed{}.

**Response:**

```
<think>
[Identifies the problem as evaluating sin(arcsin 0.4 + arcsin 0.5) · sin(arcsin 0.5
— arcsin 0.4). Defines A = arcsin(0.4) and B = arcsin(0.5), rewriting the expression
as sin(A + B) sin(B — A). Reviews basic sine addition/subtraction formulas and
considers expanding directly before looking for a better simplification strategy.]

<Parallel>
<Goal>
<Outline>1: Use sin(A + B) = sinA cosB + cosA sinB and sin(B — A) = sinB cosA —
cosB sinA, then expand the product to get a difference of squares of the form (sinB
cosA)^2 — (cosB sinA)^2.</Outline>
<Outline>2: Compute cosA from sinA = 0.4 via cosA = √(1 — sin^2 A) and compute cosB
from sinB = 0.5 via cosB = √(1 — sin^2 B), then substitute these values into the
simplified—product expression and carry out the arithmetic.</Outline>
</Goal>
<Path>
1: [Attempts expansion, realizes it's cumbersome, and switches to using the
product—to—sum identity. Applies sinα sinβ = (cos(α−β) — cos(α+β))/2 with α=A+B and
β=A−B, simplifying the product to (cos(2A) — cos(2B)) / 2.]
</Path>
<Path>
2: [Uses numerical approximations for cosA and cosB, then evaluates sin(A+B) and
sin(B—A) numerically, confirming the product is about 0.09.]
</Path>
</Parallel>

<Parallel>
<Goal>
<Outline>1: Use the product—to—sum identity sin(A+B)·sin(B—A) = [cos(2B) —
cos(2A)]/2, compute cos(2A) and cos(2B) from sinA and sinB via cos(2θ) = 1 — 2 sin²θ,
then substitute to get the exact value and check it matches the earlier numerical
estimate.</Outline>
<Outline>2: Compute A = arcsin(0.4) and B = arcsin(0.5) numerically, evaluate
sin(A+B) and sin(B—A) directly, multiply to obtain a numerical approximation, and
compare with the symbolic result.</Outline>
</Goal>
<Path>
1: [Computes cos(2A)=0.68 and cos(2B)=0.5 using the double—angle identity, plugs
into (cos(2A)−cos(2B))/2, and obtains exactly 0.09.]
</Path>
<Path>
2: [Approximates A and B numerically, computes sin(A+B) and sin(B—A), and confirms
the product closely matches 0.09.]
</Path>
</Parallel>

[Concludes that both algebraic and numerical methods agree: the value of the
expression is 0.09.]

**Final Answer**
\boxed{0.09}
</think>
```

To evaluate the expression:

$$\sin(\arcsin 0.4 + \arcsin 0.5) \cdot \sin(\arcsin 0.5 - \arcsin 0.4),$$

we begin by introducing substitutions for simplicity:
Let:
- $A = \arcsin(0.4)$, so $\sin A = 0.4$,
- $B = \arcsin(0.5)$, so $\sin B = 0.5$.

Then the expression becomes:

$$\sin(A + B) \cdot \sin(B - A).$$

**Step 1: Use Trigonometric Identity**
We apply the identity:

$$\sin(A + B) \cdot \sin(B - A) = \frac{\cos(2A) - \cos(2B)}{2}.$$

This identity is derived from the product-to-sum formulas and is particularly useful here since we can compute $\cos(2A)$ and $\cos(2B)$ using the double-angle identity:

$$\cos(2\theta) = 1 - 2\sin^2(\theta).$$

**Step 2: Compute $\cos(2A)$ and $\cos(2B)$**

$$\cos(2A) = 1 - 2\sin^2(A) = 1 - 2(0.4)^2 = 1 - 0.32 = 0.68,$$

$$\cos(2B) = 1 - 2\sin^2(B) = 1 - 2(0.5)^2 = 1 - 0.5 = 0.5.$$

**Step 3: Apply the Identity**

$$\sin(A + B) \cdot \sin(B - A) = \frac{\cos(2A) - \cos(2B)}{2} = \frac{0.68 - 0.5}{2} = \frac{0.18}{2} = 0.09.$$

**Step 4: Final Answer**
Thus, the exact value of the original expression is:

$$\boxed{0.09}$$

## E.6 Per-problem Speedup Without Filtering

We also study per-problem speedup without filtering for correctness, as shown in Figure 7. In this setting every rollout is included, so the distributions reflect the speedup that a user would observe when running each model end to end, irrespective of whether the final answer is correct. While the maximum values in the upper tail are harder to interpret, since they could capture a potential case that our model terminates earlier incorrectly while the autoregressive baseline continues reasoning and succeeds, the overall distributions remain very similar to those in Figure 5. In particular, the bulk of the mass still lies to the right of the $1.0\times$ reference line and the shape of the histograms changes only slightly. Empirically, this indicates that the measured acceleration is not primarily driven by sacrifices in accuracy, but instead reflects real reductions in token latency relative to the sequential baseline.

## E.7 Error Analysis: Repetition in Branches

We show a failure case below. Although the answer is correct, this is a failure case because the purportedly parallel threads do not provide genuinely complementary reasoning and instead largely duplicate the same computation: both compute the exponent of 5 in 42! via the standard floor-sum formula and both verify the method on smaller factorials such as 10! and 25!, so the parallel structure adds little informational value.

Moreover, the alignment between outlines and threads is not respected: rather than having one path focus on the main calculation for 42! and the other confined to checking the method on smaller examples, each path blends these tasks and performs both the primary computation and the auxiliary validations. This redundancy and mixing of roles undermines the intended division of labor across paths and illustrates why the example does not realize the benefits of structured parallel reasoning, indicating the room for improvements in the instruction following capabilities.

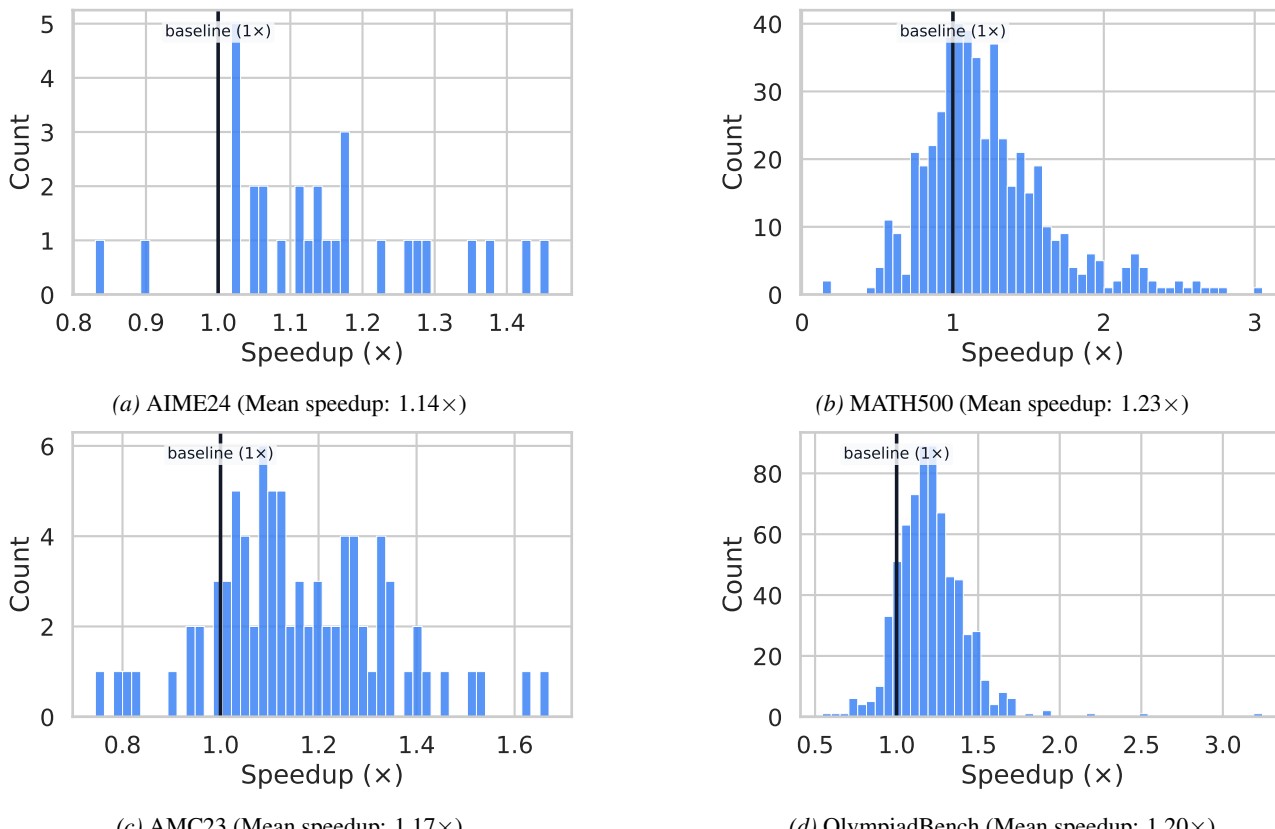

*(a)* AIME24 (Mean speedup: 1.14×)   *(b)* MATH500 (Mean speedup: 1.23×)

*(c)* AMC23 (Mean speedup: 1.17×)   *(d)* OlympiadBench (Mean speedup: 1.20×)

*Figure 7.* **Per-problem speedup distributions on math benchmarks without filtering for correctness.** The speedup is the ratio between the token latency of the autoregressive baseline and the token latency of ThreadWeaver. A vertical reference line at 1.0× marks parity with the sequential baseline. In contrast to Figure 5, these histograms include all rollouts rather than restricting to samples with correct final answers. However, taking both sets of histograms into account, we show that ThreadWeaver achieves substantial speedup on most problems no matter whether correctness filtering is performed.

---

**ThreadWeaver Example (Failure Case): Number of trailing zeros in 42!**

**Prompt:**
How many zeroes are at the end of 42! (42 factorial)? (Reminder: The number $n!$ is the product of the integers from 1 to $n$. For example, $5! = 5 \cdot 4 \cdot 3 \cdot 2 \cdot 1 = 120$.) Let's think step by step and output the final answer within \boxed{}.
**Response:**

```
<think>
[Recognizes that trailing zeros come from factors of 10 = 2·5 in the prime
factorization, and that in factorials there are many more 2s than 5s, so the number
of trailing zeros is the exponent of 5 in 42!. Plans to use the standard floor-sum
formula.]

<Parallel>
<Outlines>
<Outline>1: Use the formula for the exponent of 5 in 42!: sum floor(42/5^k) over k =
1, 2, ... until the terms are zero.</Outline>
<Outline>2: Validate the method on smaller factorials such as 10! and 25! where the
number of trailing zeros is known.</Outline>
</Outlines>
<Thread>
1: [Computes floor(42/5) = 8, floor(42/25) = 1, floor(42/125) = 0, then adds 8 + 1 +
0 = 9 and concludes that 42! has 9 trailing zeros, since 5s are the limiting factor
compared to 2s.]
```

```
</Thread>
<Thread>
2: [Applies the same floor-sum to 10! and 25!, obtaining 2 and 6 trailing zeros
respectively, which match the known values and confirm the correctness of the
counting method.]
</Thread>
</Parallel>

<Parallel>
<Outlines>
<Outline>1: Reinterpret the floor-sum as directly counting multiples of 5 and extra
factors from higher powers such as 25 among the integers from 1 to 42.</Outline>
<Outline>2: Optionally count powers of 2 in 42! to confirm that there are more than
enough 2s so that 5s indeed determine the number of trailing zeros.</Outline>
</Outlines>
<Thread>
1: [Lists multiples of 5 up to 42: 5, 10, 15, 20, 25, 30, 35, 40 (8 numbers), each
contributing at least one factor 5. Notes that 25 = 5² contributes one additional
factor 5, giving 8 + 1 = 9 factors of 5 in total, consistent with the floor-sum
calculation.]
</Thread>
<Thread>
2: [Computes the exponent of 2 in 42! as floor(42/2) + floor(42/4) + floor(42/8) +
floor(42/16) + floor(42/32) = 21 + 10 + 5 + 2 + 1 = 39, which is much larger than 9,
so the count of trailing zeros is indeed determined by the 9 factors of 5.]
</Thread>
</Parallel>

[Concludes that, since there are 9 factors of 5 and more than enough factors of 2,
42! has 9 trailing zeros.]

**Final Answer**
\boxed{9}
</think>
```

To determine how many trailing zeros are in $42!$, we look at how many times 10 divides $42!$. Since $10 = 2 \cdot 5$ and factorials contain many more factors of 2 than 5, the number of trailing zeros is the exponent of 5 in the prime factorization of $42!$.
We use the standard formula for the exponent of a prime $p$ in $n!$:

$$v_p(n!) = \sum_{k=1}^{\infty} \left\lfloor \frac{n}{p^k} \right\rfloor.$$

Here $p = 5$ and $n = 42$:

$$\left\lfloor \frac{42}{5} \right\rfloor = 8, \qquad \left\lfloor \frac{42}{25} \right\rfloor = 1, \qquad \left\lfloor \frac{42}{125} \right\rfloor = 0.$$

So the total number of factors of 5 in $42!$ is

$$8 + 1 + 0 = 9.$$

Since there are far more factors of 2 than 5 in $42!$, each of these 9 factors of 5 pairs with a factor of 2 to form a factor of 10, which contributes one trailing zero.
Therefore, the number of trailing zeros in $42!$ is

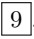 .

## F   Reasoning Trajectory Annotation Pipeline for ThreadWeaver

This section describes the multi-step pipeline that converts reasoning trajectories extracted from Qwen3-8B into a structured, parallelizable supervision signal for ThreadWeaver. Our pipeline consists of five steps:

1. Stage 1 uses GPT-5 to identify potential parallel structures within sequential reasoning traces.
2. Stage 2 rewrites these parallel structures into canonical `<Thread>` format.
3. Stage 3 performs revision against allowed and forbidden contexts to make the transition between context and each

    thread smooth.
4. Stage 4 regenerates an outline for every branch.
5. Stage 5 runs filtering to remove samples with failed rewriting or format errors to produce the final training corpus.

We describe the implementations of each step in detail.

**Step 1** In Stage 1, raw trajectories are line-numbered and assigned stable identifiers (`Lxx`). This step ingests the resulting question–answer trajectories and has the language model reconstruct the entire chain of thought into a rigorously indexed outline that reuses these line identifiers. Each reasoning trace is presented in full, the major conceptual transitions are enumerated as top-level steps with precise line spans, and aborted explorations are retained to preserve the empirical distribution of strategies. The model is further instructed to decompose these top-level steps into contiguous substeps whenever they contain multiple thought units, producing a two-level hierarchy.

After the hierarchical labeling, the model analyzes dependencies to determine which steps or blocks can run concurrently, emitting sequential and parallel structures together with short explanations. Finally, the model serializes everything (step descriptions, substeps, and parallel threads) into a normalized XML-style summary. The line-range annotations introduced at the input are preserved in this summary and are propagated verbatim through all subsequent steps.

All the tasks in this step are conducted in a multi-round conversation, and the inputs and outputs from previous tasks are visible to later tasks.

**Task 1: Main-step extraction and global indexing**. Task 1 ingests the fully line-numbered reasoning transcript, where every sentence is prefixed with immutable tags such as `L1`, `L2`, and so on. Leveraging this dense context, the model enumerates all top-level steps sequentially, captures dead ends, and records contiguous line ranges `Lxx-Lyy` for each step, thereby establishing the canonical coordinate system that downstream tasks will reuse verbatim.

---

**Task 1 Prompt for Step 1 Processing in ThreadWeaver**

```
**Task 1: Main-Step Extraction**
Analyze the given long reasoning chain for a math problem and extract every *main*
step of it. Ignore substeps|only list the top-level thinking processes or actions.

**Use the following output format:**
```
S1: [Description of step 1] (Lxx-Lyy)
S2: [Description of step 2] (Lxx-Lyy)
...
SX: [Description of step X] (Lxx-Lyy)
...
```

**Guidelines:**
- Label each top-level step consecutively (`S1`, `S2`, ...).
  - Do *not* include any sub-numbering (no `S2.1`, etc.).
- Please capture the entire thought process presented in the reasoning chain, and do
not skip any step that includes, but not limited to, the following:
  - Initial problem understanding and analysis
  - All exploration paths (both successful and unsuccessful)
  - Case studies, checks, or tests performed
  - Any \aha" or correction (re-evaluation or re-thinking) moments
  - The final reasoning that yields the solution
- Split multiple cases or scenarios into different steps. Each case should be
allocated an independent step.
- If a reasoning path is started but later abandoned because it cannot reach a
conclusion, still capture it as a distinct step.
- Please indicate the line numbers corresponding to where each main step starts and
ends in the original reasoning chain (`Lxx-Lyy`).
  - Use the format `Lxx-Lyy` to specify the range of lines for each step.
  - S1 starts at the first line and ends at the line you list in the parentheses
  (`S1: [Description of step 1] (L1-Lyy)`).
  - S(k+1) starts on the next line after Sk ends. The delta between the starting
  line number of S(k+1) and the ending line number of S(k) is always 1.
```

```
    - Ensure there is no gap between the end of one step and the start of the next.
    - The last step must end at the final line of the original reasoning chain.
- Provide a brief summary of the content in each step that is one or two sentences
long (e.g. `SX: [Description of step X]`).
- Keep the description for each step concise yet descriptive.
- A reasoning chain may contain any number of main steps. There is no limit to the
total number of steps you can identify.
```

**Task 2: Substep extraction within fixed ranges.** Task 2 reuses both the raw transcript and the Task 1 step list as shared context, slicing each `Sx` span into adjacent substeps (`Sx.1`, `Sx.2`, ... ) without gaps or overlaps. Because every line inside the parent range is still labelled with the original line numbers, the subdivision maintains identical `Lxx-Lyy` coverage while adding finer-grained anchors for later tasks.

### Task 2 Prompt for Step 1 Processing in ThreadWeaver

```
**Task 2: Substep Extraction**
Given the previous output containing all main steps of a reasoning chain, break down
each main step into substeps **only if** its content from the original reasoning
chain can be meaningfully divided into smaller thought units. You can use the line
ranges to reference the original reasoning chain and better capture the content of
each main step.

**Output format:**
```
S1: [Description of step 1] (Lxx-Lyy)
  S1.1 [Description of substep 1.1] (Lxx-Lyy)
  S1.2 [Description of substep 1.2] (Lxx-Lyy)
  ...
  S1.X [Description of substep 1.X] (Lxx-Lyy)
S2: [Description of step 2] (Lxx-Lyy)
S3: [Description of step 3] (Lxx-Lyy)
S4: [Description of step 4] (Lxx-Lyy)
  S4.1 [Description of substep 4.1] (Lxx-Lyy)
  ...
...
SX: [Description of step X] (Lxx-Lyy)
...
```

**Guidelines:**
- A substep always uses the same parent index (`X`) as its main step (e.g. if
breaking down `S2`, label `S2.1`, `S2.2`, ...).
- For each main step, capture the entire thought process presented in it, and do not
skip any substep that includes, but not limited to, the following:
  - Initial problem understanding and analysis
  - All exploration paths (both successful and unsuccessful)
  - Case studies, checks, or tests performed
  - Any \aha" or correction (re-evaluation or re-thinking) moments
  - The final reasoning that yields the solution
- Split multiple cases or scenarios into different substeps. Each case should be
allocated an independent substep.
- Provide a brief summary of the content in each substep (e.g., `S1.2 [Description
of step 1.2]`) that is one or two sentences long.
- Keep each substep description concise and focused.
- Provide the line range for each substep (`Lxx-Lyy`).
  - The format Lxx-Lyy indicates the range of lines where the substep appears.
  - SX.1 starts at the first line of SX.
  - SX.(k+1) starts on the next line after SX.k ends.
  - Ensure there is no gap between the end of one substep and the start of the next.
  - The last substep must end at the final line of the parent main step.
- Keep the main step description and line ranges (`Lxx-Lyy`) as marked in Task 1.
```

```
- A main step may contain any number of substeps. There is no limit to the total
number of substeps you can identify for a main step.
- If a main step does not contain substeps, copy over its step number, description
and line range exactly as they appear in Task 1.
- Do *not* introduce nesting deeper than 2 levels. For example, `S2.1.1` is not
allowed.
```

**Task 3: Parallelizing main steps.** Task 3 consumes the Task 1 summary and its line ranges, analyzes dependency relationships between adjacent steps, and expresses valid parallel groupings only when two steps do not rely on the same numbered lines. The reuse of the original annotations ensures that the resulting sequential and parallel blocks are traceable back to the same context without recomputing boundaries.

---

**Task 3 Prompt for Step 1 Processing in ThreadWeaver**

```
**Task 3: Parallelizing Main Steps**
Using only the **main steps** (`S1`, `S2`, ...) extracted in Task 1, first identify
all steps or contiguous step groups that can be executed in parallel without
violating logical dependencies; then rewrite them as a structured outline for
parallel execution.

**Output format:**
```
Parallel groups:
P1: Parallel(S2, S3)
P2: Parallel(Sequential(S5, S6), S7)
...
PX: Parallel(Si, Sj, Sk...)

Outline for parallel execution:
S1: [Description of step 1] (Lxx-Lyy)
P1: Parallel[Parallel reason: ...](
  S2: [Description of step 2] (Lxx-Lyy),
  S3: [Description of step 3] (Lxx-Lyy)
)
S4: [Description of step 4] (Lxx-Lyy)
P2: Parallel[Parallel reason: ...](
  Sequential(
    S5: [Description of step 5] (Lxx-Lyy),
    S6: [Description of step 6] (Lxx-Lyy)
  ),
  S7: [Description of step 7] (Lxx-Lyy)
)
...
PX: Parallel[Parallel reason: ...](
  Si: [Description of step i] (Lxx-Lyy),
  ...
)
```

**Guidelines:**
1. **Identify Parallel Groups**
  - Find groups of adjacent main steps with **no dependencies** among them.
  - Label the groups as `P1`, `P2`, ... and represent the structure of steps within
  the group using the syntax pattern as shown above.
  - Group steps Si, Sj, ... in `Parallel(Si, Sj, ...)` **if and only if** Sj does
  NOT depend on any part of Si.
  - In a parallel block, you can combine adjacent steps that depend on each other
  into a `Sequential(Sx, Sy)` block. A sequential block can be executed in parallel
  to other main steps and sequential blocks.
  - Do not create `Sequential` blocks that are not nested in any parallel block.
  Simply list the steps in order in those cases.
  - Groups must cover contiguous step ranges in the original order.
```

```
      - Interpretation of the problem should not be parallelized with the process of
      problem-solving.
      - Different ways of self-reflection are often independent of each other and thus
      parallelizable.
      - Self-reflection on an intermediate step is often parallelizable with the
      subsequent steps.
      - In conditional logic, treat the **if** branch and **else** branch as independent
      tasks and parallelize them even though their outputs cannot both occur at runtime.
      - Only include steps that can be executed in parallel with others in a parallel
      group; the parallel groups do not need to cover every step.
      - A parallel group should not consist of only one single step or one
      `Sequential(...)` block. If this is the case, do not include those steps into any
      parallel groups.

2. **Write the Structured Outline for Parallel Execution**
      - Annotate the main steps with the parallel grouping structures.
      - Keep each main step's description and line range the same as those annotated in
      Task 1 (preserve as much as you can).
      - Include **every** step exactly once, either alone, inside a sequential block, or
      inside a parallel group.
      - If a main step does not belong to any sequential block or parallel group, copy
      over its step number, description and line range exactly as they appear in Task 1.
      Include it in the outline in its original position.
      - For a sequential block, list each step in order.
      - If a parallel group contains multiple units, briefly state why its units are
      independent in the `parallel reason` bracket (e.g. `PX: Parallel[Parallel reason:
      ...]`).

- **Contiguous Blocks:** You may combine multiple adjacent steps (e.g.
`Sequential(S1, S2, ...)`) and treat them as a single unit in the parallel group
alongside another step or block (e.g. `S3`), but do **not** combine non-adjacent
steps into a block.
- **Contiguous grouping:** Each parallel group must cover a continuous sequence of
steps or blocks. In other words, you may only parallelize adjacent steps. The
occurrence of steps in parallel groups must follow their original order. For example,
P1: Parallel(S5, S8) is not allowed.
- **Strict Parallelism Only:** Build an explicit dependency graph in your analysis:
draw an edge from step A to step B if B uses A's output or insight. A group `Pi` may
include steps (or contiguous blocks) **only** if there are **no** dependency edges
between any two units.
```

**Task 4: Parallelizing substeps.** Task 4 layers on top of the substep catalog from Task 2 and the parallel structure from Task 3, examining each main step in isolation to see which adjacent substeps can proceed concurrently. Because the inputs still reference the exact `Lxx-Lyy` intervals, the model can safely bundle parallel-friendly fragments while citing the same line numbers, preserving context reuse down to the second hierarchy level.

---

**Task 4 Prompt for Step 1 Processing in ThreadWeaver**

```
**Task 4: Parallelizing Substeps**
Examining the outline you extracted in Task 3, for any main steps that are not in a
`Parallel` block, refer to its **substeps** you extracted in Task 2 if there is any
(e.g. `S1.1`, `S1.2`, ...). Among these substeps, identify all substeps or contiguous
substep groups that can be executed in parallel without violating logical
dependencies. Then rewrite the structured parallel execution outline to include both
the parallel structures of the main steps and those of the substeps you just
identified.

**Output format:**
```
Parallel groups of substeps:
* Parallel(Sequential(S1.1, S1.2), S1.3)
* Parallel(S4.2, S4.3)
```

```
* ...
* Parallel(...)

Outline for parallel execution:
S1: [Description of step 1] (Lxx-Lyy)
  P-sub: Parallel[Parallel reason: ...](
    Sequential(
      S1.1: [Description of substep 1.1] (Lxx-Lyy),
      S1.2: [Description of substep 1.2] (Lxx-Lyy)
    ),
    S1.3: [Description of substep 1.3] (Lxx-Lyy)
  )
P1: Parallel[Parallel reason: ...](
  S2: [Description of step 2] (Lxx-Lyy),
  S3: [Description of step 3] (Lxx-Lyy)
)
S4: [Description of step 4] (Lxx-Lyy)
  S4.1: [Description of substep 4.1] (Lxx-Lyy)
  P-sub: Parallel[Parallel reason: ...](
    S4.2: [Description of substep 4.2] (Lxx-Lyy),
    S4.3: [Description of substep 4.3] (Lxx-Lyy)
  )
P2: Parallel[Parallel reason: ...](
  Sequential(
    S5: [Description of step 5] (Lxx-Lyy),
    S6: [Description of step 6] (Lxx-Lyy)
  ),
  S7: [Description of step 7] (Lxx-Lyy)
)
...
PX: Parallel[Parallel reason: ...](
  Si: [Description of step i] (Lxx-Lyy),
  ...
)
```

**Guidelines:**
1. **Identify Parallel Groups**
   – Only parallelize the substeps whose main steps are not in a parallel block.
   – Find sets of adjacent substeps with no dependencies among them.
   – Label the groups of substeps `P-sub`, and list their substep members (e.g.
   `Parallel(Sequential(S1.1, S1.2), S1.3)`, `Parallel(S4.2, S4.3)`).

2. **Write the Structured Outline for Parallel Execution**
   – Annotate the main steps and substeps with the parallel group structures.
   – Keep the parallel group structure, numbering, description and line ranges of the
   main steps the same as those annotated in Task 3 (preserve as much as you can).
   – Keep each substep's numbering, description and line range the same as those
   annotated in Task 2 (preserve as much as you can).
   – For a sequential block, list the description of each substep in order.
   – If a parallel group contains multiple units, briefly state why its units are
   independent in the `parallel reason` bracket (e.g. `PX: Parallel[parallel reason:
   ...]`)
   – If a substep does not belong to any parallel or sequential group, copy over its
   step number, description and line range exactly as they appear in Task 2. Include
   it in the outline in its original position.

– **Coverage:** For each main step you choose to annotate, include **every** substep
exactly once, either alone or inside a parallel group.
– **Contiguous Blocks:** In a parallel block, you may combine adjacent substeps that
depend on each other into a contiguous block (e.g. `Sequential(S2.1, S2.2)`) and
treat them as a single unit in the parallel group alongside another substep or block
(e.g. `S2.3`). Do **not** combine non-adjacent substeps into a block.

```
- Do not create `Sequential` blocks that are not nested in any parallel block.
Simply list the substeps in order in those cases.
- Only group substeps corresponding to the same main step in a single-level parallel
or sequential block (e.g. Parallel(S2.2, S3.1) or Sequential(S2.1, S2.2, S3.1) is
not allowed).
- **Strict Parallelism Only:** Build an explicit dependency graph in your analysis:
draw an edge from substep A to substep B if B uses A's output or insight. A group
`Pi` may include steps (or contiguous blocks) **only** if there are **no**
dependency edges between any two units.
- In conditional logic, treat the **if** branch and **else** branch as independent
tasks and parallelize them even though their outputs cannot both occur at runtime.
- Contiguous grouping only: Each parallel group must cover a continuous sequence of
steps or blocks. In other words, you may only parallelize adjacent steps or substeps.
The occurrence of steps or substeps in parallel groups must follow their original
order. For example, P1: Parallel(S2.2, S3.1) is not allowed.
```

**Task 5: XML-style serialization with ranges.** Task 5 repackages everything derived so far—main steps, substeps, and parallel groupings—into the prescribed XML-like outline. Each `<Thread>` and `<Parallel>` element explicitly repeats the inherited `Lxx-Lyy` spans, so the serialized structure is a faithful rendering of the earlier outputs that downstream tooling can align with either the raw transcript or the intermediate tables.

---

**Task 5 Prompt for Step 1 Processing in ThreadWeaver**

```
**Task 5: Annotate the Overall Structured Outline with Line Range for Each Parallel
Block**
According to the conversation above, output the overall parallel execution structure
of the long reasoning chain with the main steps, substeps and the parallel
structures clearly defined in a simple XML format.

In addition, include the line ranges for each parallel block and its threads.

**Output format:**
```
S1: [Description of step 1]
  <Parallel>[Parallel reason: ...]
  <Thread>
  S1.1: [Description of substep 1.1]
  S1.2: [Description of substep 1.2]
  </Thread> (range: Lxx-Lyy)
  <Thread>
  S1.3: [Description of substep 1.3]
  </Thread> (range: Lxx-Lyy)
  </Parallel> (range: Lxx-Lyy)
<Parallel>[Parallel reason: ...]
<Thread>
S2: [Description of step 2]
</Thread> (range: Lxx-Lyy)
<Thread>
S3: [Description of step 3]
</Thread> (range: Lxx-Lyy)
</Parallel> (range: Lxx-Lyy)
S4: [Description of step 4]
  S4.1: [Description of substep S4.1]
  <Parallel> [Parallel reason: ...]
  ...
  </Parallel> (range: Lxx-Lyy)
<Parallel> [Parallel reason: ...]
<Thread>
S5: [Description of step 5]
S6: [Description of step 6]
</Thread> (range: Lxx-Lyy)
...
```

```
</Parallel> (range: Lxx-Lyy)
..
```

**Guidelines:**
- Format the output **strictly** according to the output format specified above.
**The only XML tags allowed are <Parallel> and <Thread>.** A <Parallel> </Parallel>
block should consist of multiple <Thread>'s. <Thread> can only exist inside
<Parallel>.
- A <Parallel> block may form at either the main step level, or the substep level.
- Ensure **there are no `<Parallel>` tags nested inside one another.** Do not create
structures like <Parallel> ... <Parallel> ... </Parallel> ... </Parallel>.
- **Max depth of substep nesting is 2.** Ensure there is no step number beyond Sx
and Sx.y. For example, `Sx.y.z` is not allowed.
- **Do not add additional tags for sequential structures.** If an XML node's
children are purely sequential, list them in order.
- **Tag parallel blocks.** Wrap parallelizable units in `<Parallel>`...`</Parallel>`.
Wrap each parallelizable unit (including sequential blocks and (sub)steps) in
`<Thread>`...`</Thread>`. List the sequential steps and substeps in order inside each
`<Thread>` block.
- Provide the line range for each <Thread> block after the closing tag (`</Thread>
(range: Lxx-Lyy)`).
- Provide the line range for each <Parallel> block after the closing tag
(`</Parallel> (range: Lxx-Lyy)`).
- Keep the description for every step and substep as annotated in Task 1 and 2
(preserve as much as you can).
- **Do not create significantly imbalanced parallel blocks.** If most of an XML
node's children are sequential and only a small number of them can run in parallel,
you may leave the group untagged or only create a parallel block using the children
that are genuinely parallel.
- Do not start your output with "```xml". Indentation (i.e., spaces before Sx.y)
indicates substeps, not XML structures; do not add tags other than <Parallel> and
<Thread> around them.
- **Group parallelizable sets.** You may bundle several independent units into a
single `<Parallel>` block when they share no dependencies.
```

**Task 6: Consistency verification across contexts.** Task 6 revisits the complete context (the original transcript plus all prior task outputs) to double-check that every instruction was satisfied. By reusing the same line numbering and dependency notes, it validates that no steps were skipped, that parallel reasoning is justified, and that references respect the allowed-versus-forbidden context policy introduced earlier in the prompt.

---

**Task 6 Prompt for Step 1 Processing in ThreadWeaver**

```
**Task 6: Double-check Every Parallel Block and Refine the Structured Outline**
Please check and revise the structured outline for parallel execution you generated
in Task 5 by ensuring the following:
- **Coverage & Contiguity**: Make sure the line range (Lxx-Lyy) of each thread
exactly spans from the first included (sub)step to the last. Make sure the line
range (Lxx-Lyy) of each parallelizable block exactly spans from the first included
thread to the last.
- **Parallelism Validity**: Verify that there are no direct or indirect dependencies
among the children of each parallel block. Specifically, confirm that each thread in
the block does not depend on *any* of the threads before it in the same block. Add a
note after the closing tag of each thread (`</Thread>`) indicating the result of
this check.
  - If the check passes, add the note `check passed` and confirm the reason
  (`</Thread> [range: Lxx-Lyy, check passed: reason]`).
```

```
   - If the check fails (i.e. dependency is detected within the parallel block),
     refactor the problematic block into Sequential(...) or separate groups, and update
     the line ranges accordingly. If you cannot determine how to refactor, leave a note
     stating`check failed` along with the reason (`</Thread> [range: Lxx-Lyy, check
     failed: reason]`).
 - **Output Format**: Keep the outline format as specified in Task 5. Only the tags
`<Parallel>` and `<Thread>` are allowed, and `<Thread>` tags can only appear inside
`<Parallel>` blocks.

**Output format:**
```
S1: [Description of step 1]
  <Parallel>[Parallel reason: ...]
  <Thread>
  S1.1: [Description of substep 1.1]
  S1.2: [Description of substep 1.2]
  </Thread> (range: Lxx-Lyy, check passed: first thread)
  <Thread>
  S1.3: [Description of substep 1.3]
  </Thread> (range: Lxx-Lyy, check passed: S1.3 has no dependencies on S1.1 and
  S1.2)
  </Parallel> (range: Lxx-Lyy)
<Parallel>[Parallel reason: ...]
<Thread>
S2: [Description of step 2]
</Thread> (range: Lxx-Lyy, check passed: first thread)
<Thread>
S3: [Description of step 3]
</Thread> (range: Lxx-Lyy, check failed: S3 depends on S2)
</Parallel> (range: Lxx-Lyy)
S4: [Description of step 4]
  S4.1: [Description of substep S4.1]
  <Parallel> [Parallel reason: ...]
  ...
  </Parallel> (range: Lxx-Lyy)
<Parallel> [Parallel reason: ...]
<Thread>
S5: [Description of step 5]
S6: [Description of step 6]
</Thread> (range: Lxx-Lyy, check passed: first thread)
...
</Parallel> (range: Lxx-Lyy)
...
```

**Step 2** The second step is a deterministic restructuring pass that converts the free-form sequences emitted above into canonical parallel markup. It sweeps over every annotated trajectory, infers the boundaries of each reasoning branch, and wraps the contiguous spans inside explicit thread containers that sit beneath the appropriate parallel block. During this pass the tool enforces ordering constraints, ensures that adjacent steps remain contiguous after threading, and drops in the dependency notes that certify why two segments may or may not execute simultaneously. Crucially, the script preserves the inline `Lxx` prefixes that Step 1 injected into every reasoning line, so the downstream prompts can continue to reference exactly the same line-number ranges even after the text is reorganized into `<Thread>` blocks. Because the logic consists solely of programmatic rewrites plus structural validation, no additional prompting is necessary.

**Step 3** Next, each candidate trajectory undergoes a linguistic compliance review that compares the text under inspection against immediate predecessors and lateral sections. The reviewer model receives a slice of "allowed" context plus a sibling "forbidden" context, scans the target passage for implicit carry-overs, and proposes fine-grained edits to excise any leakage from the forbidden branch. The output takes the form of minimal substring replacements in JSON, which allows automated application while keeping every other token untouched. Running this step across all paths eliminates hidden cross-references, making each branch self-contained before outline regeneration.

## Prompt for Step 3 Processing in ThreadWeaver

In this task, your goal is to check the English language of a given text segment against two other text segments from the same article:
* The "allowed context" segment comes from the immediate preceding section of the text segment under review. The text segment under review may reference content from this previous section.
* The "forbidden context" segment is from a section **parallel** to the text segment under review. The text segment under review must not reference any content from this parallel section.

Specifically, you will be provided with the following:
1. Allowed Context | contains information the text under review may reference.
2. Forbidden Context | contains information the text under review must not reference.
3. Text Under Review | the text segment to be checked.

Your task is to decide whether the text under review refers to any information in the Forbidden Context.
If it does, output a JSON array of substring replacements that remove or rewrite the referencing language in the text under review, following the output format specified below.
If it does not, output an empty array: `[]`.
Output the raw JSON only. Do not enclose the output in "```json...```".

Output format:
```
[
  {
    "src": "<the substring to be removed or rewritten (exact match, properly
    escaped)>",
    "dest": "<the replacement text which removes the reference to the Forbidden
    Context (use \"\" to delete)>"
  },
  ...
]
```

**Guidelines:**
* Replace the minimal substrings needed to eliminate the reference to the Forbidden Context while keeping the text under review grammatical.
* Escape quotes and newlines in JSON (use `\"` and `\n`).
* A proposed replacement will affect all occurrences of the "src" field  in the text under review. If you do not intend to modify all the occurrences, make the substring in "src" more specific.
* Do not introduce new information from the Forbidden Context in your replacements.
* References to the Allowed Context are permitted. Do not remove any references to the Allowed Context.
* Do not add explanations or code fences.
* Do not output any text outside the JSON.

Special cases to watch for:
* Remove or rewrite any occurrences of:
  - Anaphora/Deictics: Phrases like "as above," "as before," "same as earlier,"
  "that result," "those values," which implicitly point to the Forbidden Context.
  Keep these when they refer to the Allowed Context.
  - Implicit carry-over: "unchanged from the previous section," "as we computed,"
  when the "previous" information is from the Forbidden Context.
  - Shorthand references: Symbols, variable names, or equation labels defined only
  in the Forbidden Context (e.g., "that z," "the value from (3)").
  - Cross-references: "see previous," "see above," footnote markers, or inline
  citations that point to the Forbidden Context.
  - Paraphrased pointers: Rewrites that still rely on the Forbidden Context even
  without verbatim overlap.

```
* Overlapping matches: When multiple candidate substrings overlap, prefer one
replacement of the longest meaningful phrase rather than many small ones.
* Case/whitespace variants: Handle differences in capitalization or spacing exactly
by matching the literal substring present in the path.
* Multiline fragments: If the referencing text to be replaced spans lines, include
`\n` in src and dest as needed.
* Grammar preservation: If removing a piece of referencing text  leaves dangling
punctuation or grammar issues, include the surrounding words/punctuation in "src" to
keep the sentence clean.
* Nothing salvageable: If an entire clause or sentence is a forbidden reference
(e.g., "Which is the same as before."), delete it by setting "dest": "".

Allowed Context:
...
[5 lines before the <Parallel> block]

Forbidden Context:
[First 5 lines at the beginning of the prior <Thread> block]
...
[Last 5 lines at the end of the prior <Thread> block]

Text Under Review:
[Full text of the <Thread> block to check]
```

**Step 4** Once the sanitized paths are in place, a fresh model call rebuilds the planning scaffolds that will ultimately supervise downstream learners. Using only the contextual window that precedes each parallel block, the model drafts concise yet actionable outlines for every branch, restating the tactical moves a mathematician would plan before executing the proof. Each outline mirrors the ordering of the paths, isolates the reasoning intent without revealing conclusions, and keeps the different branches independent so later steps can treat them as interchangeable schedules. Because the outlines are derived strictly from the provided context, they serve as high-quality teacher plans for training planning modules or evaluating interpretability.

---

**Prompt for Step 4 Processing in ThreadWeaver**

```
In this task, your goal is to generate outlines for some mathematical reasoning
procedures. Specifically, you will be given
1. A coherent math reasoning trace segment tagged with parallel structures
(`<Parallel> ... <Thread> ... </Thread> ... </Parallel>`) that indicate reasoning
steps which can be executed in parallel.
2. A group of placeholders placed at the beginning of the segment
```
<Outlines>
<Outline>1: [Outline to be generated for path 1 below]</Outline>
<Outline>2: [Outline to be generated for path 2 below]</Outline>
...
<Outline>X: [Outline to be generated for path X below]</Outline>
</Outlines>
```
3. A segment of text immediately preceding the target reasoning trace segment
(`Context:`) to serve as the context for deriving the outline.

Using only the provided context (`Context:`), generate the outline for each <Thread>
in the <Parallel> segment. Place the generated outline in the markers "[Outline to
be generated for path X below]".

**Output format:**
```
<Outlines>
<Outline>1: [Outline to be generated for path 1 below]</Outline>
<Outline>2: [Outline to be generated for path 2 below]</Outline>
...
```

```
<Outline>X: [Outline to be generated for path X below]</Outline>
<Outlines>
```

**Guidelines:**
* Imagine that a mathematician is thinking and using the outlines to structure and guide their thinking process in each corresponding path (`<Outline>X:... <Thread>X:...`).
* Frame the outlines so the mathematician could logically plan them based solely on the context, without access to the content in the parallel segment or any additional information.
* Make the outline directly and easily derivable from the context { neither too abstract nor too specific { and strictly aligned with the order of the paths.
* Each outline should capture the gists and logical structure of its path without excessive details.
* Thinking in each path is done independently. Ensure that the outlines remain distinct and do not depend on one another.
* Do not reveal exact calculations, numerical results, or the final conclusion from its path in the outline. Instead, indicate the steps/methods/directions that lead to them.
* Do not introduce a substantial amount of reasoning in the outlines.
* Avoid phrases like "conclude that ...", "rule out that ...", or anything that cannot be directly derived from the context without significant reasoning. Do not frame the outline as if the conclusion is already known.
* A reader should be able to write the path from the outline plus the context alone with some reasoning effort, without additional information.
* Keep the outline concise and clear: one sentence for short content; at most two sentences for longer content.
* Output plain text within each outline block: no bullet points, numbering, headings, or extra commentary.
* Do not start your output with "```xml". Start your output with <Outlines> and end your output with </Outlines>. The only allowed tags in a <Outlines> block are <Outline> tags.

**Step 5** The final step is a comprehensive filtering pass that operates on the model-generated artifacts. It traverses every outline file, enforces formatting contracts, checks that parallel tags are balanced, and discards trajectories that still violate structural constraints after the preceding corrections, ensuring that only high-quality, well-structured demonstrations enter the supervised learning pipeline.

## G   Discussion: Reward Hacking

A potential concern in RL with parallel rollout is *reward hacking*, where the model artificially inflates the acceleration reward by generating numerous redundant branches. In our experiments, we did not see clear evidence of this behavior. Our model benefits from strong reasoning priors established during supervised fine-tuning on high-quality, parallel-annotated trajectories, which guide it towards meaningful decomposition rather than excessive branching. Moreover, our RL rollout framework enforces a strict 40k total token limit per trajectory, which also discourages excessive branching (e.g., enumerating thousands of trivial cases). The acceleration ratio reward itself is intentionally conservative, by capping the reward and assigning a small scaling factor to it, so that correctness dominates the overall learning signal.

One solution for this type of reward hacking, which may happen in future search or application, is to apply a length penalty on the token latency. While seemingly more effective than acceleration ratio, it introduces normalization challenges, as the magnitude of the token counts is not naturally comparable to correctness rewards, and poorly normalized penalties can degrade performance by over-penalizing legitimate long reasoning traces. In contrast, the acceleration ratio is normalized relative to each trajectory's structure (i.e., comparing the token latency with total number of tokens) and bounded, maintaining stability and fairness across tasks. For future extensions, more complicated length-based penalties may be applied to the token latency to better account for the potential reward hacking, but in our experiments, the clipped acceleration reward proved sufficient to keep acceleration high without compromising reasoning quality.

# H Future Directions

## H.1 Topology-aware Parallelized Reasoning

The policy used in ThreadWeaver is *topology-agnostic*: it decides how many `<Thread>`s to spawn based on problem structure, not on available accelerators or interconnect characteristics. This limits scalability and efficiency. In contrast to CPU multithreading, where runtimes expose core counts and schedulers adapt thread pools, the model currently has no visibility into the number of GPUs, memory limits, or link bandwidth/latency. As a result, it may under-utilize large clusters or over-subscribe small ones, incurring overheads without reducing the critical path latency.

A natural extension is to condition the policy on a *hardware profile* at inference time, e.g., a concise header with (GPUs, communication bandwidth and latency, topology map). The orchestrator (Section 2.2) can pass this profile via lightweight control tokens along with optional SLOs. A *topology-aware* policy would then adapt its fork–join plan, number of `<Thread>`s, granularity of subproblems, and join cadence, to the exposed resources, much like manually tuned tensor/pipeline parallelism, but learned and input-conditional.

Training can mirror this via *domain randomization* over hardware profiles and SLOs, coupled with a latency- and cost-aware variant of P-GRPO reward function. The acceleration bonus (Section 4.2) can be normalized by an estimate of the topology's maximum attainable concurrency, while penalties account for scheduling overheads and stragglers. Ultimately, this points to a resource-conditioned planner that maps problem DAGs to device graphs, automatically discovering the best parallelization strategy for the hardware at hand.

## H.2 Parallelizing Interactions Beyond Reasoning

Our framework can be interpreted as a multi-agent system, where the model autonomously determines when to branch its reasoning into multiple concurrent threads. Each thread functions as an independent agent that explores a subproblem and later synchronizes with others through structured aggregation. This implicit coordination mechanism allows the model to decompose reasoning tasks dynamically, but the same principle could extend far beyond text-based reasoning.

An exciting future direction is to generalize parallelization from internal reasoning to interactive decision-making in an environment. In such a setting, multiple agents could act concurrently within a shared environment, each pursuing complementary objectives or subtasks, towards one goal. For instance, in software engineering applications, distinct agents might analyze separate components of a complex system, refactor interconnected modules, and verify consistency through communication protocols. Similar frameworks could emerge in domains such as scientific discovery, robotics, or large-scale planning, where agents must balance autonomy with coordination.

Enabling such forms of parallelized interaction introduces new challenges in communication and synchronization. Simple message passing may be insufficient when agents share mutable contexts or overlapping goals. Mechanisms such as hierarchical coordination, role-based delegation, or dynamic resource locking could provide more robust foundations for scalable multi-agent reasoning. Exploring how these interaction protocols can be learned end-to-end rather than manually specified remains an open and promising direction for future research.

