# OpenReview forum: "ThreadWeaver: Adaptive Threading for Efficient Parallel Reasoning in Language Models"
_ICML.cc/2026/Conference — ICML 2026 spotlight_

### Official Review · Reviewer_EzyM · 2026-03-02

**Soundness:** 4
**Presentation:** 4
**Significance:** 3
**Originality:** 3
**Overall Recommendation:** 5
**Confidence:** 4

**Summary:**

This work presents an adaptive framework that enables LLMs to dynamically fork reasoning into concurrent threads and rejoin them to finalize a single, coherent solution.
The system is built upon three core innovations: a two-stage parallel trajectory generator that produces high-quality data for supervised fine-tuning, a trie-based co-design for seamless deployment on standard autoregressive engines, and a Parallelization-aware GRPO (P-GRPO) reinforcement learning framework.
By utilizing additional parallel compute to decompose tasks, the model significantly reduces the latency of the critical reasoning path. While this approach can result in higher total token counts than traditional sequential decoding due to structural overhead and redundancy, it effectively trades compute for faster wall-clock speed.
ThreadWeaver trained on top of Qwen3-8B achieves good scores on math benchmarks (79.9% on AIME24 and 71.9% on average) while delivering up to 1.53× speedup in token latency.

**Compliance With Llm Reviewing Policy:**

Affirmed.

**Final Justification:**

My concerns have been adequately addressed. I raised my score from 4 to 5.

**Key Questions For Authors:**

## Questions for Authors

1. **Regarding the 1.22× vs 1.14× discrepancy:** Could you clarify if the 1.22× "Average Speedup" is an arithmetic mean of per-problem ratios? If so, why is this a more representative metric than the ratio of the mean token latencies (1.14×), and how do you ensure that a few highly parallelizable outliers are not overstating the global efficiency of the framework?
2. **Regarding stragglers and resource idling:** In Figure 3, the orchestrator must wait for all threads to reach the `</Thread>` token before proceeding. How does the system handle cases where threads are highly imbalanced in length? Specifically, do you have a mechanism to dynamically re-allocate GPU resources to other tasks if one thread finishes significantly earlier than its siblings?
3. **Regarding the limits of acceleration:** Given the constraints of Amdahl's Law, have you identified a theoretical upper bound for the parallelization of math reasoning? Furthermore, how does the P-GRPO reward function prevent "reward hacking," where the model might spawn redundant or trivial threads just to increase the acceleration ratio without actually solving the problem faster?
4. **Regarding hardware sensitivity:** Since the policy is currently topology-agnostic, how would the performance and fork-join decisions change if the number of available GPUs was varied (e.g., 1 vs 8)? How do you envision a "hardware-conditioned" policy affecting the model's willingness to branch into more or fewer threads?

**Limitations:**

yes

**Strengths And Weaknesses:**

## Strengths

* **System-Agnostic Originality:** The work provides a novel combination of trie-based prefix trees and standard autoregressive engines. This allows the model to perform adaptive parallel reasoning without requiring specialized inference kernels or modifications to attention masks at runtime, distinguishing it from prior literature that requires custom system modifications.
* **Soundness of Reinforcement Learning Framework:** The introduction of Parallelization-aware GRPO (P-GRPO) is technically robust. It provides a mathematically justified method for broadcasting trajectory-level advantages across parallel threads, addressing the complex credit assignment challenge in branched trajectories without relying on ad-hoc heuristics.
* **Significance of Performance Parity:** This work highlights that parallelized reasoning can maintain the depth of long-chain-of-thought models. While prior methods often suffered from accuracy degradation, ThreadWeaver matches cutting-edge sequential models (e.g., 79.9% on AIME24), establishing a new Pareto frontier for high-latency reasoning tasks.

## Weaknesses

* **Inconsistency in Efficiency Metrics:** The paper reports an "Average Speedup" of 1.22× across benchmarks in Table 1. However, it also notes that the mean critical-path length across the same datasets was reduced from 15.1k to 13.2k tokens, which translates to a lower aggregate speedup of ~1.14×. My understanding is that the 1.22× figure likely represents an average of per-problem ratios, which can be skewed by outliers, whereas the ratio of the means (1.14×) is a more conservative and accurate reflection of total compute-time reduction across a full workload.
* **System Overhead and the "Straggler" Effect:** The authors utilize token latency (the length of the critical path) as the primary proxy for inference speed. The paper admits that real-world wall-clock latency (1.14× speedup) is lower than these token-level gains (1.23× speedup) due to system overheads like repeated API calls and KV cache prefilling. My understanding is that in a high-concurrency environment, if one parallel thread runs significantly longer than others, the system must wait for this "straggler" to finish before joining, potentially leaving allocated GPU resources idle.
* **Task-Type Limitations and Amdahl's Law:** The paper describes ThreadWeaver as an adaptive framework where the model "spontaneously" decides when to fork reasoning. However, the results show that speedup is highly dependent on a task's "divide-and-conquer" nature, with some benchmarks like AIME25 showing only a negligible 1.03× speedup. My understanding is that the framework is strictly bound by Amdahl’s Law; for tasks that are intrinsically sequential, the overhead of generating `<Outline>` tokens and redundant reasoning across threads could actually result in a performance floor that limits the method's generalizability.
* **Lack of Hardware-Awareness in Policy:** The authors acknowledge that the current policy is topology-agnostic, meaning it forks reasoning based on problem structure without visibility into the available hardware. My understanding is that because the model does not know if it is running on one GPU or a large cluster, it might over-subscribe small clusters or under-utilize large ones. This suggests that the reported speedups are highly sensitive to the specific experimental configuration and may not scale predictably across different hardware environments.

---

> ### Author Rebuttal · Authors · 2026-03-31
>
> Thank you for the thoughtful review and your acknowledgement of our novel design, solid RL formulation, and the strong results on the Pareto-frontier. We answer your questions and address your concerns below:
>
> **[W1][Q1] Interpreting aggregate speedup metrics and sensitivity to outliers**
>
> This is a valuable point. The two speedup metrics capture different notions of representativeness. The 1.22x figure is the mean of per-problem speedups, reflecting the expected gain when sampling a dataset uniformly. By contrast, 15.1k / 13.2k = 1.14x is a ratio of mean token latencies, which weights benchmarks with longer trajectories more heavily. For example, an additional 1k tokens may have limited relative impact on long-horizon tasks, but can substantially affect shorter ones.
>
> To mitigate sensitivity to such extremes while retaining the benefits of both views, we also compute a trimmed ratio-of-means, excluding benchmarks with the highest and lowest mean latencies. This yields an intermediate estimate of ~1.20x. We believe this provides a more balanced summary of the overall acceleration achieved by ThreadWeaver. We will clarify the distinction between these metrics and report them with interpretation.
>
> **[W2][Q2] Straggler effect, scheduling, and system-level efficiency**
>
> Our runtime follows a fork-join pattern, so each join waits for the slowest thread. This is why we optimize critical-path length rather than total generated tokens. In our vLLM-based serving setup, each thread is issued as a backend call, and resources freed by completed requests can be reassigned dynamically to other pending threads or tasks. As noted, realized speedup is smaller than the reduction in token latency, and further overhead reduction could help close this gap.
>
> That said, we view the most relevant deployment setting for ThreadWeaver as latency-sensitive serving in high-capacity, autoscaled environments, where assigning extra parallel compute to a small number of difficult queries is often practical and has limited effect on other requests. In these settings, the system-level cost of stragglers is less severe than in a fully saturated cluster.
>
> More broadly, for hard reasoning workloads, the value of latency reduction is often super-linear. This is reflected in practice by premium [fast mode tiers](https://developers.openai.com/codex/speed) offered by OpenAI Codex, where speedups (1.5x) can support disproportionately higher pricing (2x). ThreadWeaver is also designed for this regime and is complementary to such system-level optimizations.
>
> **[W3][Q3] Amdahl’s law and limits of parallel speedup**
>
> ThreadWeaver, like any parallelization method, is fundamentally constrained by Amdahl’s law: the achievable speedup is bounded by the sequential portion of the reasoning trace. Our claim is therefore not universal acceleration, but that adaptive intra-trajectory parallelism can substantially shorten the critical path when tasks contain meaningful decomposable structure.
>
> In fact, the average parallelization ratio is relatively high in our current model output trajectories (65.7% in AIME2024), suggesting that such structure is common in the regime we study. Furthermore, increased RL could potentially optimize the model to obtain chains-of-thought with even higher parallelization ratio. At the same time, a high parallelization ratio does not directly translate into proportional wall-clock speedup, since gains are still limited by the remaining sequential segments, branch imbalance, and system overhead. We will clarify in the revision that ThreadWeaver should be viewed as improving the speed-accuracy Pareto frontier, rather than overcoming the fundamental limits imposed by sequential computation.
>
> **[Q3] Reward hacking**
>
> We also considered the possibility of reward hacking, where the model could artificially increase the acceleration reward by spawning redundant, low-value, or trivial branches. In practice, we did not observe clear evidence of this behavior. Our acceleration reward is correctness-gated, clipped, and deliberately small relative to the main correctness reward, so the model does not benefit much from parallelization unless it still solves the problem correctly.
>
> In addition, the rollout is subject to a strict total token budget (40k), which discourages excessive branching and makes wasteful decompositions costly, as the parallelization reward is much smaller than the main correctness reward. We present additional discussions in Section G in our appendix.
>
> **[W4][Q4] Lack of hardware awareness**
>
> Good point! This is an important future direction. Our current work is primarily an algorithmic contribution rather than an infrastructure or hardware-aware scheduling contribution. A natural next step is a hardware-conditioned policy that adapts branching decisions to the available resources.
>
> We hope our responses have addressed your concerns and help clarify the intent and the practical implications of our work.

---

> > ### Author Rebuttal · Reviewer_EzyM · 2026-04-01
> >
> > My concerns have been adequately addressed. I will raise my score.

---

### Official Review · Reviewer_2nz6 · 2026-03-11

**Soundness:** 2
**Presentation:** 3
**Significance:** 3
**Originality:** 3
**Overall Recommendation:** 4
**Confidence:** 2

**Summary:**

This is an inference method for reasoning model to reduce inference latency while preserving accuracy. the paper can spawn parallel branches that solve independent sub-tasks and later merge their results, thereby shortening the critical reasoning path. To enable this capability, the author introduce several technical components. First, they design a structured reasoning format with control tokens that represent parallel blocks and threads. Second, they propose a trie-based training and inference co-design that converts parallel trajectories into independent training sequences. Third, they introduce a reinforcement learning method called P-GRPO, which optimizes both correctness and acceleration through a reward that combines accuracy with parallelization efficiency.

**Compliance With Llm Reviewing Policy:**

Affirmed.

**Final Justification:**

The rebuttal has addressed my concerns, I will keep my score.

**Key Questions For Authors:**

The experiments focus primarily on mathematical reasoning. Have the authors evaluated ThreadWeaver on domains such as code generation or agentic tasks?

Launching multiple reasoning threads for each request consumes additional batch capacity on the gpu. Could the authors report end-to-end latency and throughput under saturated batch conditions? This would help clarify whether the proposed parallelization improves real-world system performance or mainly shifts compute from inter-request parallelism to intra-request parallelism.

**Limitations:**

yes

**Strengths And Weaknesses:**

Strengths:
The design of P-GRPO is well justified. The authors provide a clear explanation of why standard GRPO variance normalization can lead to unstable optimization in the presence of multi-term rewards, and they support their design choice with empirical ablations showing improved stability when using mean-centered advantages instead. This analysis strengthens the credibility of the proposed training procedure and helps clarify how reward signals interact in the parallel reasoning setting. In addition, the inference architecture is carefully designed to remain compatible with standard autoregressive inference engines. By avoiding modifications to model architectures or serving systems, the approach can be deployed on existing inference stacks such as vLLM-style engines, which significantly lowers the practical barrier to adoption. The overall system design demonstrates a thoughtful integration of training, inference, and system considerations.

Weaknesses:
The method assumes that parallel reasoning threads can be generated and executed independently, which may not hold for many reasoning tasks where intermediate steps have strong dependencies on one another.  Additionally, the empirical gains remain moderate. A concern relates to how the evaluation measures latency. In practical deployments, spawning multiple branches per problem may reduce the number of requests that can be served, potentially decreasing overall throughput and increasing end-to-end latency under realistic workloads. Therefore, evaluating only single-request latency may overestimate the practical benefit of the approach. A more informative evaluation would measure end-to-end latency under fully utilized batch conditions, where the batch is saturated with multiple requests, to better understand the trade-off between parallel reasoning and system throughput.

---

> ### Author Rebuttal · Authors · 2026-03-31
>
> Thank you for the detailed review and for highlighting the practical value of our approach!
>
> **[W1] [Q1] Applicability to tasks with dependent intermediate steps / generalization beyond math**
>
> We agree that not all reasoning tasks can be easily parallelized, especially when later steps depend strongly on earlier intermediate results. However, many important tasks do naturally contain parallelizable structures.
>
> In particular, many high-value workloads, including agentic coding and deep research, contain partially independent subproblems, candidate paths, or local analyses that can be explored concurrently even when the overall task is not fully parallel. To provide evidence of this beyond the math domain, we ran our data pipeline on 100 SWEBench-Verified trajectories generated by Kimi K2.5 with access to a Linux sandbox. We find that 47.12% of the model output is parallelizable, and exploiting this structure yields a 1.15x speedup in token latency.
>
> For example, in `sympy__sympy-18189`, the trajectory identifies a bug in SymPy where results depend on the order of symbols, then splits into two independent threads: one inspects the implementation to find the cause, and the other reproduces and validates the behavior from the outside. Because those threads do not depend on each other, they can run in parallel and then merge cleanly into a small fix plus verification. Importantly, these results are obtained without any task-specific prompt engineering for software engineering. The current acceleration is likely limited in part by branch imbalance in our pipeline, and we expect that task-specific prompt engineering could produce more balanced branches and larger gains.
>
>
> **[W2] Empirical efficiency gains**
>
> We agree that the acceleration (e.g., ~1.2x on average with up to ~3.5x in some cases) may appear moderate. However, the economic value of latency reduction is often super-linear in practice. For example, OpenAI Codex’s pricing [increases 2x with a 1.5x speedup](https://developers.openai.com/codex/speed) when a user turns on the “fast mode”. As a result, even moderate acceleration can already translate into meaningful economic value. We therefore believe the current gains are already practically meaningful, and that further algorithmic and systems advances can substantially expand this speed-accuracy frontier.
>
> At the same time, to our knowledge, ThreadWeaver is the first adaptive parallel reasoning method that enables strong long-CoT thinking models to tackle competition-level math while maintaining parity with a capacity-matched cutting-edge used sequential RL baseline (Qwen3-8B). We view this result as an initial proof of feasibility and the identification of a new Pareto point. We believe further advances, such as multi-level parallelism and additional systems optimizations could unlock even larger gains along the speed-accuracy frontier.
>
> **[W3] [Q2] Latency evaluation under saturated batch conditions**
>
> Parallel reasoning methods, including prior work (APR, Multiverse, etc.) as well as our approach, are not primarily designed to reduce the latency in a highly batched setting under a limited GPU budget. Instead, as highlighted in Sec 1 of our paper (L80-84), the target setting is latency-sensitive hard queries in high-capacity, autoscaled serving environments, where assigning extra compute to a small fraction of valuable requests often has little impact on the overall throughput of each API call of the system. Such setting is practical and common, as frontier models (e.g., GPT, Claude, and Gemini) are often served this way and exposed with APIs. In such settings, our method enables an effect analogous to the “fast mode” available in OpenAI Codex. Note that our method can be applied together with acceleration from the system side (e.g., “fast mode”), enabling complementary speed improvements.
>
> At the same time, we acknowledge that some use cases prioritize lower cost over latency. To accommodate this, ThreadWeaver supports a hybrid mode: it can operate in parallel mode when latency is critical, or revert to standard autoregressive decoding when throughput is the main concern.

---

> > ### Author Rebuttal · Reviewer_2nz6 · 2026-04-02
> >
> > The rebuttal resolved my concerns, and I will keep my score unchanged.

---

### Official Review · Reviewer_hZGB · 2026-03-13

**Soundness:** 3
**Presentation:** 3
**Significance:** 4
**Originality:** 3
**Overall Recommendation:** 5
**Confidence:** 3

**Summary:**

This paper presents ThreadWeaver, a framework for adaptive parallel reasoning that decomposes a single long reasoning trajectory into parallel threads without modifying the underlying autoregressive inference engine; the approach combines structured parallel-trajectory supervision, trie-based train–inference alignment, and a parallelization-aware RL objective (P-GRPO). On six math benchmarks, ThreadWeaver matches the sequential RL baseline in accuracy (71.9% vs. 72.2%) while reducing average token latency from 15.1k to 13.2k (1.22× speedup).

**Compliance With Llm Reviewing Policy:**

Affirmed.

**Final Justification:**

The rebuttal has addressed my main concerns, and I will keep my score.

**Key Questions For Authors:**

1. Fairness of comparison. What happens if the sequential baseline is given the same additional test-time compute budget?
2. Generalization beyond math. Does the method extend to non-mathematical reasoning tasks?
3. Dependence on external data construction. Would the method still work without the GPT-5 rewriting pipeline? The cold-start set contains only 959 examples, derived from a five-stage GPT-5 rewriting process over 53k sequential trajectories. If a weaker teacher were used, or if such an expensive annotation pipeline were unavailable, how much would performance degrade? This is directly relevant to reproducibility and general applicability.
4. What is actually being improved? Is the method improving reasoning itself, or mainly improving decomposition into parallel subproblems? The ablations suggest that both self-training and parallel rollout are important, but it remains unclear whether the gains come primarily from better problem solving or from better parallelization. Is there any quantitative analysis of decomposition quality itself, such as thread independence, join quality, or outline quality?

**Limitations:**

yes

**Strengths And Weaknesses:**

Strengths
1. The paper has strong practical engineering value.
2. The method is compatible with standard inference engines, which makes deployment more plausible.
3. The trie-based train/inference alignment is a technically meaningful design choice rather than a superficial implementation detail.

Weaknesses
1. For tasks that cannot be decomposed into parallel subproblems, the proposed approach is likely to have limited benefit.
2. The evaluation scope is limited, as experiments are conducted only on mathematical reasoning tasks.
3. The paper considers only single-level parallelization, while multi-level nested parallel reasoning is not sufficiently discussed.
4. The data construction pipeline depends heavily on a strong external model and a large number of prompts, making it difficult to isolate and demonstrate the effect of ThreadWeaver itself.
5. The reward design still raises potential concerns regarding reward hacking and hardware mismatch.

---

> ### Author Rebuttal · Authors · 2026-03-31
>
> Thank you for the insightful and encouraging review and positive assessment! We address each point below.
>
> **[W1] Parallelizability and decomposability of tasks**
>
> We agree that ThreadWeaver is most beneficial when tasks contain decomposable structure. That said, many tasks with high economic value, such as agentic coding and deep research, do have such a structure. For example, to evaluate whether our method applies to agentic coding, we ran our data pipeline on 100 SWEBench-Verified trajectories generated by Kimi K2.5 with access to a Linux sandbox. We find that 47.12% of the model output is parallelizable, and that exploiting this parallelism yields a 1.15x speedup in token latency. These findings suggest that our task decomposition approach generalizes beyond math reasoning. Importantly, these gains are achieved without domain-specific prompt engineering for software engineering tasks, leaving substantial room for further improvement through task-specific optimization.
>
> **[W2] [Q2] Applicability beyond math tasks**
>
> We focus on math because it provides realistic long-horizon reasoning with reliable verifiable rewards. As mentioned in the response to [W1], our framework is not math-specific and is promising when applied to additional tasks such as agentic coding.
>
> **[W3] Multi-level parallelization**
>
> As discussed in the paper, we intentionally focus on single-level parallelization as a clean and well-controlled initial setting, and find that it already covers many use cases while delivering strong performance. In principle, extending to multi-level or nested parallelism is straightforward: the same fork-join state machine can be applied recursively within each thread, along with the corresponding trie-based training and ancestor-only masking.
>
> **[W4] [Q3] Role of external teacher model for data curation**
>
> The external teacher is used only for a small cold start with 959 SFT examples to teach the parallel behavior. Most of the signal instead comes from self-training, which produces 17k model-generated trajectories. The amount of external supervision from 959 samples is comparable to prior and concurrent work on parallel reasoning, such as Multiverse and Parallel-R1, and modest relative to large SFT pipelines like DeepSeek-Distilled-Qwen, which uses [800k samples](https://huggingface.co/deepseek-ai/DeepSeek-R1-Distill-Qwen-7B) from DeepSeek-R1. In addition, as open-source models improve, we expect the same model being trained could also serve as the data processor, removing the need for an external model.
>
> **[W5] Reward robustness and hardware mismatch**
>
> We discussed the possibility of reward hacking in Section G in the appendix. The acceleration reward is correctness-gated, clipped, and small relative to the correctness reward, so correctness remains dominant and we do not see reward hacking in practice.
>
> For hardware mismatch, we treat token latency as a proxy and also report wall-clock gains. A topology-aware or hardware-aware system is a natural future direction, which we discuss in Section H in the appendix.
>
> **[Q1] Sequential baseline with increased test-time compute**
>
> Our comparison focuses on the setting where extra parallel hardware is available and latency matters. Giving our current sequential baseline more test-time budget makes it a stronger baseline in terms of performance, but would degrade in terms of latency, which is the main quantity ThreadWeaver is designed to improve.
>
> Furthermore, we tested a longer-context sequential baseline (60k vs. 40k). It improved only slightly, from 78.3% to 78.5% accuracy, while increasing average token use from 19.4k to 20.1k. Even linearly scaled to our model’s 21.1k total token budget, the sequential baseline still falls short of our 79.9% accuracy. Not to mention our model achieves this while using only 16.9k average tokens on the critical path.
>
> **[Q4] Source of gains: decomposition vs. raw reasoning improvements**
>
> The main improvement is better decomposition / critical-path reduction while largely preserving reasoning quality, rather than a large standalone jump in raw reasoning ability. This is justified by the fact that 1) our sequential baseline is trained with the same set of prompts in RL, and 2) our rewriting pipeline for SFT cold-start data generation is mostly rearranging existing statements in the reasoning traces that the base model itself is able to generate.
>
> To analyze task decomposition quality, we perform an LLM-as-a-judge analysis on our reasoning traces. We use `gpt-4.1-mini` to score AIME24 trajectories across five dimensions on 30 prompts with 8 samples each. Overall, the judge assigns high scores across all five dimensions.
>
> | | Correct Trajectories | Incorrect Trajectories | Overall |
> | - | - | - | - |
> | Decomposition meaningfulness | 4.69 | 4.14 | 4.57 |
> | Path independence | 4.73 | 4.19 | 4.61 |
> | Outline alignment | 4.79 | 4.04 | 4.63 |
> | Join quality | 4.93 | 3.81 | 4.69 |
> | Raw reasoning quality | 4.93 | 3.44 | 4.60 |

---

> > ### Author Rebuttal · Reviewer_hZGB · 2026-04-03
> >
> > I thank the authors for their rebuttal. They have provided extensive clarifications addressing the concerns I raised about the paper, and my questions have now been fully resolved. And I will keep my score.

---

### Official Review · Reviewer_dTCg · 2026-03-13

**Soundness:** 3
**Presentation:** 3
**Significance:** 3
**Originality:** 3
**Overall Recommendation:** 5
**Confidence:** 3

**Summary:**

This paper proposes ThreadWeaver, a training method for eliciting parallel reasoning. Unlike previous attempts, ThreadWeaver matches the performance of sequential reasoning but at reduced latency.

**Compliance With Llm Reviewing Policy:**

Affirmed.

**Final Justification:**

I will keep my positive assessment and recommend accept.

**Key Questions For Authors:**

See weaknesses.

**Limitations:**

Yes

**Strengths And Weaknesses:**

Strengths:
1. The method is effective at preserving the performance of sequential reasoning, but at reduced latency.
2. The trie-based loss computation is neat and helps to eliminate the training-inference mismatch that could otherwise jeopardize training stability. Compared to naive techniques (e.g., constructing separate sequences at each fork), this approach is far more efficient.
3. The ablations are informative.

Weaknesses:
1. The evaluation is limited to math, and it's unclear if the method would generalize to less verifiable tasks (e.g., subagents for software engineering).
2. Intuitively, I'd think that longer sequential traces have more opportunities to parallelize. The results suggest the opposite: On average, AIME2025 requires the most tokens but little speedup, while MinervaMath has the fewest tokens but the highest speedup. This raises doubts about how well the method could generalize to really long (e.g., 10-hour) tasks.

---

> ### Author Rebuttal · Authors · 2026-03-31
>
> Thank you for the thoughtful review and for highlighting the method’s effectiveness, simplicity, and efficiency! We answer your questions below:
>
> **[W1] Generalization beyond mathematical reasoning tasks**
>
> We chose math as our main focus because it provides reliable verifiable rewards without leveraging external tools or environments, allowing a clean test of whether adaptive parallel reasoning can match strong sequential reasoning while reducing latency. However, the core components of ThreadWeaver are domain-agnostic, and we would like to provide some evidence that our method can be naturally extended to agentic domains such as agentic coding.
>
> Specifically, we ran our data pipeline through 100 SWEBench-Verified trajectories generated by Kimi K2.5 with access to a Linux sandbox. We found that a large portion (47.12%) of the model output is parallelizable. For these trajectories, we observe that parallelization leads to 1.15x acceleration in terms of token latency. Although we did not train a new SWE agent model due to the limited rebuttal period, these results suggest that our task decomposition approach can generalize beyond math. Notably, these results are obtained without domain-specific prompt engineering for SWE tasks, and we expect that task-specific optimization could lead to even larger gains.
>
> **[W2] When and why longer reasoning traces yield greater speedup**
>
> In our view, speedup depends on both problem decomposability and model familiarity with the task distribution. Our SFT and RL are smaller scale compared to Qwen’s official post-training pipeline, and we hypothesize that the prompt dataset that we used, Polaris-53k, has less coverage on AIME2025, compared to other datasets. In this situation, the model tends to 1) make more mistakes during task decomposition, and 2) fall back to the sequential reasoning pattern learned during Qwen’s official post-training stage, leading to less parallelization. By contrast, upon analysis we found that MinervaMath dataset contains more opportunities for independent sub-computations or verification, which yields larger critical-path reduction despite shorter average traces. We will clarify this interpretation in the revision.
>
> To investigate if the longer traces yield a larger speedup, we get rid of the dataset choice as the confounder. We found that for the same dataset (AIME2024), the model achieves 1.14x acceleration for trajectories with 0-10k tokens and 1.46x acceleration for trajectories with 30k-40k tokens (note that we use a 40k context window). This indicates that under the same dataset distribution, longer samples do lead to higher acceleration when parallelized.

---

> > ### Author Rebuttal · Reviewer_dTCg · 2026-04-04
> >
> > I appreciate the detailed responses. I find them convincing. I will keep my positive assessment.

---

### Decision · Program_Chairs · 2026-04-30

**Decision:**

Accept (spotlight)

**Comment:**

This paper proposes ThreadWeaver, a framework for adaptive parallel reasoning in LLMs that decomposes sequential chain-of-thought into concurrent threads via a fork-join mechanism, combining parallel trajectory data generation, trie-based training-inference alignment, and a parallelization-aware RL objective (P-GRPO). On six math benchmarks, it matches sequential reasoning accuracy (79.9% on AIME24) while achieving up to 1.53x speedup in token latency. The paper establishes a new Pareto frontier between accuracy and latency for reasoning models, which is a meaningful contribution to the field. All reviewers marking concerns as fully resolved after rebuttal.

Recommendation: Accept